# Pervasive isoform-specific translational regulation via alternative transcription start sites in mammals

Xi Wang[1,†], Jingyi Hou[1,†], Claudia Quedenau[1] & Wei Chen[1,2,*]

## Abstract

Transcription initiated at alternative sites can produce mRNA isoforms with different 5′UTRs, which are potentially subjected to differential translational regulation. However, the prevalence of such isoform-specific translational control across mammalian genomes is currently unknown. By combining polysome profiling with high-throughput mRNA 5′ end sequencing, we directly measured the translational status of mRNA isoforms with distinct start sites. Among 9,951 genes expressed in mouse fibroblasts, we identified 4,153 showed significant initiation at multiple sites, of which 745 genes exhibited significant isoform-divergent translation. Systematic analyses of the isoform-specific translation revealed that isoforms with longer 5′UTRs tended to translate less efficiently. Further investigation of *cis*-elements within 5′UTRs not only provided novel insights into the regulation by known sequence features, but also led to the discovery of novel regulatory sequence motifs. Quantitative models integrating all these features explained over half of the variance in the observed isoform-divergent translation. Overall, our study demonstrated the extensive translational regulation by usage of alternative transcription start sites and offered comprehensive understanding of translational regulation by diverse sequence features embedded in 5′UTRs.

**Keywords** alternative transcription start sites; *cis*-regulatory elements; isoform-divergent translation; translational regulation

**Subject Categories** Chromatin, Epigenetics, Genomics & Functional Genomics; Genome-Scale & Integrative Biology; Protein Biosynthesis & Quality Control

**Mol Syst Biol. (2016) 12: 875**

## Introduction

Eukaryotic gene expression is a complex process orchestrated by multiple regulatory steps, of which transcription and translation are the two most important ones. Recent genome-wide studies have demonstrated that both processes play critical roles in determining cellular protein abundance (Schwanhäusser *et al*, 2011; Marguerat *et al*, 2012; Li *et al*, 2014; Jovanovic *et al*, 2015; Li & Biggin, 2015; Liu *et al*, 2016). In contrast to prokaryotes, where transcription and translation are closely coupled, in eukaryotes the two procedures are spatially and temporally separated. It remains underexplored whether and to what extent the regulation between eukaryotic transcription and translation could be coordinated.

Eukaryotic transcription outputs, that is, mRNA transcripts, consist of not only the coding sequences (CDS) that dictate the encoded peptide sequences, but also 5′ and 3′ untranslated regions (UTRs). Various *cis*-elements that are functionally implicated in translational regulation are known to be embedded in 5′UTRs. As a textbook example, iron response elements in the 5′UTR regulate the translation of ferritin mRNA according to the cellular iron level. In addition, upstream open reading frames (uORFs) are known to repress translation of the main ORFs (Mueller & Hinnebusch, 1986; Matsui *et al*, 2007; Calvo, 2009) and *in vitro* analyses have demonstrated that stable RNA secondary structures near 5′ cap could block translation initiation (Kozak, 1989). Therefore, revisiting the possible coordination between eukaryotic transcription and translation, one likely scenario is through the usage of alternative promoters and thereby assembling divergent translational regulatory *cis*-elements in distinct 5′UTRs. Indeed, previous studies in yeast have shown that around two hundred yeast genes express isoforms with different 5′UTRs, many of which show diverse translational profiles (Arribere & Gilbert, 2013). Both *in vitro* and *in vivo* analyses have demonstrated that different 5′UTR sequences derived from the same yeast genes can lead to large difference in translational efficiency (TE) (Rojas-Duran & Gilbert, 2012).

Compared to unicellular yeast, promoter architecture in mammals displays much higher complexity and transcription could initiate from much broader genomic regions (Lenhard *et al*, 2012; Li *et al*, 2015). Genome-wide analyses have demonstrated that around half of human and mouse genes have multiple promoters (Cooper *et al*, 2006; Kimura *et al*, 2006; Baek *et al*, 2007). The most recent transcription start site (TSS) survey from the FANTOM consortium, which includes 573 human primary cell samples, 152 human tissues, and 250 human cancer cell lines, has revealed that on average there are four TSSs per gene, and moreover, that TSS

1   Laboratory for Functional Genomics and Systems Biology, Berlin Institute for Medical Systems Biology, Max-Delbrück-Centrum für Molekulare Medizin, Berlin, Germany
2   Department of Biology, South University of Science and Technology of China, Shenzhen, Guangdong, China
    *Corresponding author. Tel: +86 75588018449; E-mails: wei.chen@mdc-berlin.de; chenw@sustc.edu.cn
    †These authors contributed equally to this work

usage is highly dynamic and regulated in a cell type-specific manner (Forrest *et al*, 2014). It is therefore conceivable that alternative TSSs could substantially expand the 5′UTR repertoire, conferring great potential for differential translational regulation. Indeed, individual examples have shown that alternative TSSs can drastically alter the 5′UTR structure and thereby result in enhanced or diminished protein synthesis rate (Pozner *et al*, 2000; Blaschke *et al*, 2003; Courtois *et al*, 2003). Such TSS switches are usually of functional significance and frequently associated with pathologic phenotypes (Arrick *et al*, 1991; Sobczak & Krzyzosiak, 2002). One well-known example is tumor suppressor gene *BRCA1*, which has two isoforms with distinct 5′UTRs due to its alternative TSSs. The efficiently translated shorter isoform is expressed in both cancerous and non-cancerous breast tissues, whereas the translationally inactive longer 5′UTR isoform is only expressed in tumor tissues, leading to decreased BRCA1 protein abundance observed in sporadic breast and ovarian cancers (Sobczak & Krzyzosiak, 2002). Very recently, Doudna and colleague attempted to determine mRNA isoform-specific translational regulation by combining polysome profiling and RNA sequencing. They determined isoform-specific translational status by calculating isoform abundance in different fractions using Cufflinks suite based on Ensembl gene annotations and found that properties of 3′UTRs predominated over 5′UTRs as the driving force behind isoform-specific translational regulation (Floor & Doudna, 2016). Compared to previous studies surveying the effect on polysome association by only either alterative splicing or differential usage of 3′UTRs (Spies *et al*, 2013; Sterne-Weiler *et al*, 2013), their study could in principle more comprehensively assess the relative contribution of diverse features from different regions along the whole transcripts. However, precise estimation of isoform abundance based solely on short-read RNA-seq data, as applied in their study, is still an unresolved challenge (Angelini *et al*, 2014). In addition, their study was based on the rather incomplete TSS annotation and therefore would unavoidably underestimate the role of 5′UTR diversity in translational regulation.

Here, to directly characterize the global impact of TSS diversity on translational regulation, we combined polysome profiling with high-throughput mRNA 5′ end sequencing to measure the translational status of mRNA isoforms with distinct TSSs (TSS isoforms). In murine fibroblasts, we identified a total of 22,357 TSSs derived from 10,875 protein-coding genes, about 54% of which were not annotated in either RefSeq or Ensembl databases. Among 4,153 genes showing significant initiation at multiple TSSs, we identified 745 genes exhibiting significant TE difference between their alternative TSS isoforms and found that longer isoforms were more frequently associated with lower TE. By correlating the observed isoform-specific TE with the presence/absence of various sequence features, we demonstrated the global impact of several regulatory elements, such as uORFs, cap-adjacent stable RNA secondary structures, and 5′-terminal oligopyrimidine (5′ TOP). In addition, we also identified several novel sequence motifs that can affect translation activity and validated the effect of two using reporter systems. Finally, we constructed a quantitative model to assess the combinatory effect of different features identified in this study, which could explain over 50% of the variance of the TE difference observed between alternative TSS isoforms.

# Results

## Genome-wide assessment of translational efficiency associated with distinct TSS isoforms

Polysome profiling, in which mRNAs bound by different number of ribosomes are separated into multiple fractions on a sucrose gradient through velocity sedimentation, is a well-established and widely used method to assess the *in vivo* translational status of mRNAs (Arava *et al*, 2003; Arribere & Gilbert, 2013; Spies *et al*, 2013). In order to assess the TE of distinct TSS isoforms, we combined polysome profiling with mRNA 5′ end sequencing. In short, we collected RNAs from seven gradient fractions and quantitatively profiled 5′ ends of mRNA transcripts in each fraction by adapting the cap-trapping approach (Carninci *et al*, 1996) with Illumina sequencing (Materials and Methods). To ensure that the sequencing read counts from different fractions can be used to estimate their relative abundance of the same transcripts even though the total mRNA content varied across different fractions, we added to each fraction the same amount of *Drosophila melanogaster* total RNA and used the read counts derived from the spike-in RNA for across-fraction normalization. Thereafter, we quantified translational status of distinct TSS isoforms by calculating the averaged number of associated ribosomes (Fig 1A).

We applied this method in a population of exponentially growing non-synchronized NIH3T3 mouse fibroblasts. In each of the two biological replicates, we sequenced the 5′ ends of RNAs collected from the seven fractions. On average, we obtained 46.2 million paired-end reads per fraction, and about 78% of post-ncRNA filtering reads were uniquely aligned to the mouse genome and used in the following analyses (Materials and Methods; Table EV1). In total, we identified 22,357 TSSs derived from 10,875 protein-coding genes (Materials and Methods). Among these, 17,033 (76.2%) TSSs were mapped within gross 5′UTRs of 9,951 protein-coding genes, including both annotated 5′UTRs ($n = 13,705$) and 1 kb upstream of annotated TSSs (Up 1 kb; $n = 3,328$) (Fig 1B). The remaining TSSs were located either in CDS ($n = 1,934$), downstream introns ($n = 3,216$), or 3′UTRs ($n = 174$). Although some of them may represent the residue retention of cDNAs with incomplete 5′ ends, many may lead to mRNAs encoding N-terminal truncated protein isoforms or even non-coding transcripts (see Discussion). Since this study focused on the quantitative effect of alternative 5′UTRs on TE, to avoid other complicating factors, we used only the TSSs lying within the gross 5′UTRs for further analyses.

For each of these 17,033 TSSs, we estimated its relative TE by calculating the averaged numbers of its associated ribosomes based on their normalized sequencing read counts from different fractions (Materials and Methods). The results correlated very well between the two biological replicates, demonstrating the high reproducibility of our approach (Fig EV1A). Hierarchical clustering of the sequencing data from different fractions recapitulated the gradient order (Fig EV1B), indicating the accurate polysome profiling. By and large, mRNAs encoding longer ORFs were bound by more ribosomes, reflecting that CDS length limits the number of associated ribosomes (Spearman $\rho = 0.53$; Fig EV2A). Interestingly, the mRNAs with shorter ORF ($\leq 450$ nt) appeared to be enriched in 80S monosome fraction (Fig EV2B), agreeing to the recently observed active monosome translation of short ORF in yeast (Arava *et al*,

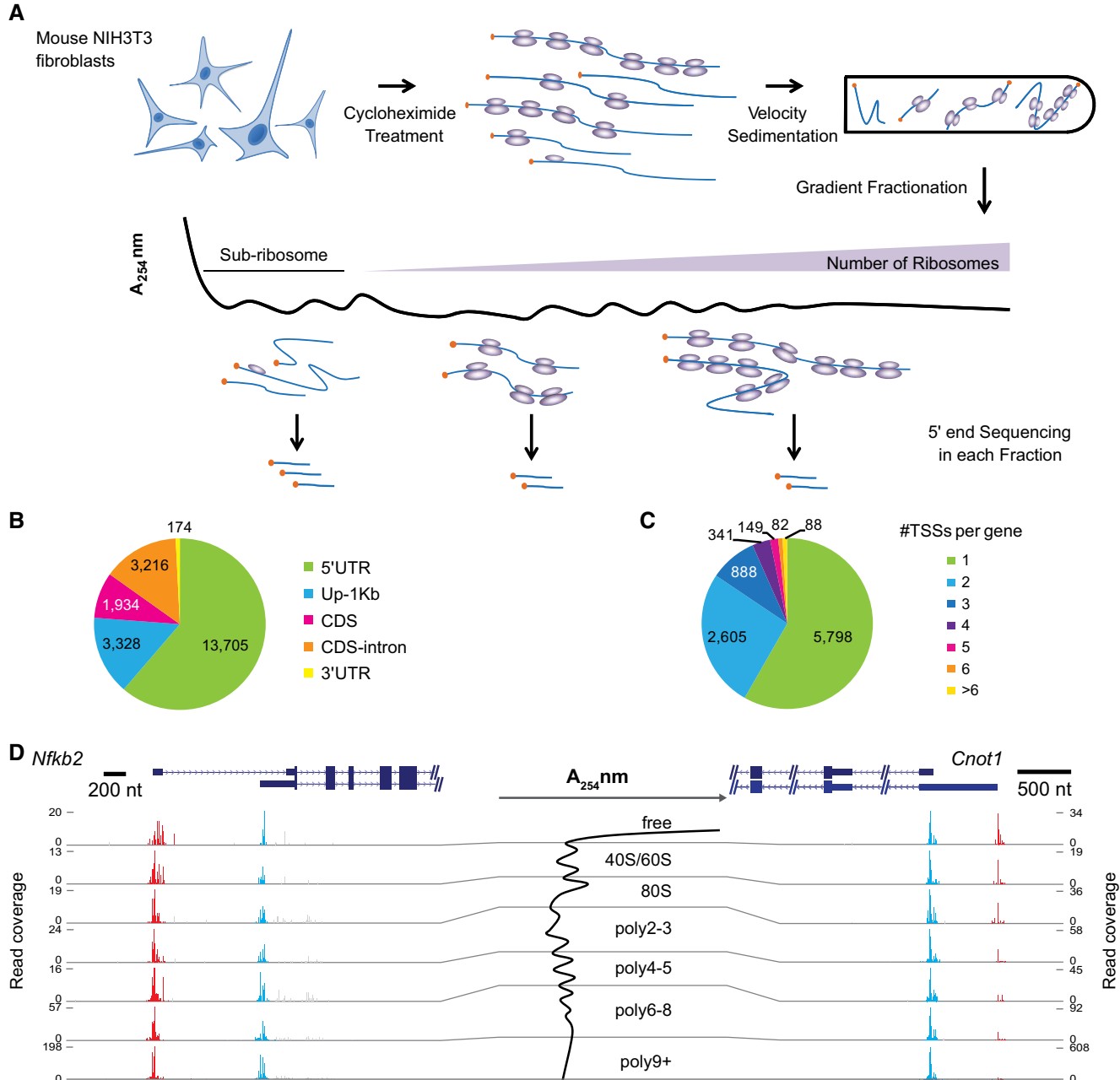

**Figure 1. Experimental scheme, TSS discovery, and examples of isoform-specific translational efficiency (TE).**

A  Experimental scheme. RNAs were collected from seven gradient fractions and the 5′ ends of RNA transcripts were quantitatively profiled in each fraction using an adapted cap-trapping approach.

B  Pie chart showing the distribution of TSSs identified in this study in different regions of protein-coding genes. The majority of TSSs were derived from gross 5′UTRs, including annotated 5′UTRs and 1 kb upstream of the annotated TSSs (Up-1 kb).

C  Pie chart showing the number of TSSs in the gross 5′UTRs per protein-coding gene. Out of the 9,951 genes with at least one TSS detected, 4,153 (41.7%) expressed multiple TSSs.

D  Two examples were shown to demonstrate the impact of alternative TSSs on TE. Cumulative reads along each gene from the seven gradient fractions (shown in the middle) were plotted under the gene structure. While the two alternative TSSs from gene *Nfkb2* resulted in no difference in TE, the two from gene *Cnot1* led to substantial TE difference. Please note the range of read coverage varied across fractions. Red and blue bars represented sequencing reads mapped within distal and proximal TSSs, respectively; gray bars represented reads mapped outside of the identified TSSs. The description of the two genes can be found in Table EV3.

2003; Heyer & Moore, 2016). To further confirm that our approach could quantitatively capture translational status, we compared the TE values obtained from our polysome profiling to those based on

ribosome footprinting (Eichhorn *et al*, 2014) and protein synthesis rate based on proteomics measurement (Schwanhäusser *et al*, 2011). To compute TE for each gene, we combined our data for

alternative TSS isoforms and then normalized against its ORF length. These TE values correlated well with those based on proteomics (Spearman $\rho$ = 0.46, Fig EV2C) and even better with those derived from ribosome footprinting (Spearman $\rho$ = 0.57, Fig EV2D).

## Alternative TSSs lead to differential TE in 745 out of 4,153 multi-TSS genes

Out of 9,951 genes with at least one TSS detected in the gross 5′UTRs, 4,153 (41.7%) genes showed significant initiation at multiple TSSs (Fig 1C). Whereas the genes with single TSS tended to express higher and were enriched in those encoding proteins with housekeeping functions such as translation, genes with multi-TSS were enriched in regulatory pathways (Fig EV3A).

To investigate the impact of alternative TSS usage on translational regulation, for each of these 4,153 genes with multiple TSSs, we compared TE fold changes between pairs of its alternative TSS isoforms. Of 13,118 pairwise comparisons, the 5–95$^{th}$ percentile interval of the absolute values of log2-transformed fold changes spanned a range between 0.02 and 2.2. As shown in Fig 1D, while a value close to zero (*Nfkb2*, log2-FC = 0.06) indicates there is nearly no difference in TE between the two TSS isoforms, a value largely deviated from zero (*Cnot1*, log2-FC = 0.91) represents substantial TE difference. To check whether the result could be predominantly affected by data collected from one fraction, we performed leave-one-fraction-out analysis, in which each of the seven fractions was left out and the TE divergence between isoforms was calculated based on the remaining six fractions. As shown in Fig EV2E, the result from leave-one-fraction-out analysis showed high correlation with the original result based on data from all fractions.

To further assess the significance of the estimated TE differences, we applied a bootstrapping which could account for the estimation uncertainty associated with small read counts derived from less abundant TSS isoforms in certain fractions. For each of the 1,000 bootstrapping replicates, log2-transformed TE fold changes between isoform pairs were calculated in the same manner as in the real data, and altogether yielded a bootstrap distribution, which was then summarized with a mean and a standard deviation (Materials and Methods). The larger the bootstrap mean deviates from zero, the larger the TE diverges between the isoform pairs. By contrast, lower bootstrap standard deviation gives more confidence in the estimated TE difference. Based on the bootstrap mean and standard deviation, the statistical significance was then determined for each comparison (Fig 2A; Table EV2). After applying a threshold of Benjamini–Hochberg adjusted *P*-value < 0.01 and TE divergence > 1.5 in both replicates (FDR = 5.2%), we identified 745 genes exhibiting significant TE difference between 1,618 pairs of TSS isoforms. By and large, the dominant isoforms with higher abundance were also translated in higher efficiency. Such trend became more obvious when the difference in the alternative isoform level increased (Fig EV3B). Interestingly, at gene level, highly expressed genes were also translated more efficiently than the lowly expressed ones (Fig EV3C). Collectively, these observations suggested that mammalian cells might coordinate transcription and translation to reduce the energy consumption in the optimal growth condition as analyzed in this study.

To verify the observed isoform-specific TE, we randomly chose four genes with significant TE difference between their TSS isoforms for validation. Using an independent cap-capturing strategy based on specific cap-dependent linker ligation (Materials and Methods), we amplified the 5′ end complete cDNA products from RNA extracted from non-ribosomal fraction and polysomal fraction separately. All of the cDNA products were of the size consistent with the corresponding TSSs (Fig 2B). More importantly, the ratio of relative abundance of TSS isoforms between non-ribosomal and polysomal fractions agreed to that determined by our global approach (Fig EV4).

To further examine whether the sequence difference of alternative 5′UTRs is able to drive the observed TE divergence, we used an *in vivo* reporter system to compare the TE of a *Renilla* luminescent reporter gene led by the 5′UTR sequences derived from paired alternative TSS isoforms identified in eight genes (Materials and Methods). As shown in Fig 2C, seven out of the eight pairs showed significant differential TE biased toward the same isoforms as observed in our global analysis. Notably, the 5′UTR sequence from *Ndufb11* shorter isoform resulted in eleven times higher TE than that from the longer one, demonstrating alternative 5′UTR sequences can confer significant contribution to translational regulation.

## Isoforms with longer 5′UTR tend to have lower translational efficiency

To understand the ways in which TSS isoforms differentially affect TE, we first sought to check for the global effect of 5′UTR length and observed two interesting trends based on 6,536 pairwise comparisons between alternative isoforms with unambiguously determined 5′UTRs (Materials and Methods). First, as shown in Fig 3A, the larger the length difference between the two isoforms, the higher the fraction associated with significant TE divergence, indicating that longer divergent sequences might contain more regulatory elements exclusively used by the long isoforms. Second, more interestingly, when plotting the relative TE for long and short TSS isoforms derived from the same genes, we evidenced a global tendency that longer isoforms were associated with lower TE (Fig 3B). Among the 1,025 isoform pairs with significant TE difference, nearly 80% (814) showed a longer 5′UTR lower TE bias (Fig 3B). Such trend became more prominent with the increase of 5′UTR length difference between TSS isoforms (Fig 3A), suggesting that 5′UTR sequences in general comprised of more translational repressive elements than enhancing ones.

## Upstream translation starting at AUG negatively affects the main ORF translation

Next, we aimed to characterize the sequence features in the 5′UTRs that could account for the observed TE divergence between alternative TSS isoforms. Given the effect of 5′UTR length observed above, in the following analyses, we always matched the 5′UTR length difference between the two groups of comparison.

Upstream ORFs (uORFs) and upstream AUGs (uAUGs, without in-frame stop codons in the 5′UTRs) have been reported to negatively affect TE of the main ORFs (Mueller & Hinnebusch, 1986; Matsui *et al*, 2007; Calvo, 2009). To check whether the presence of

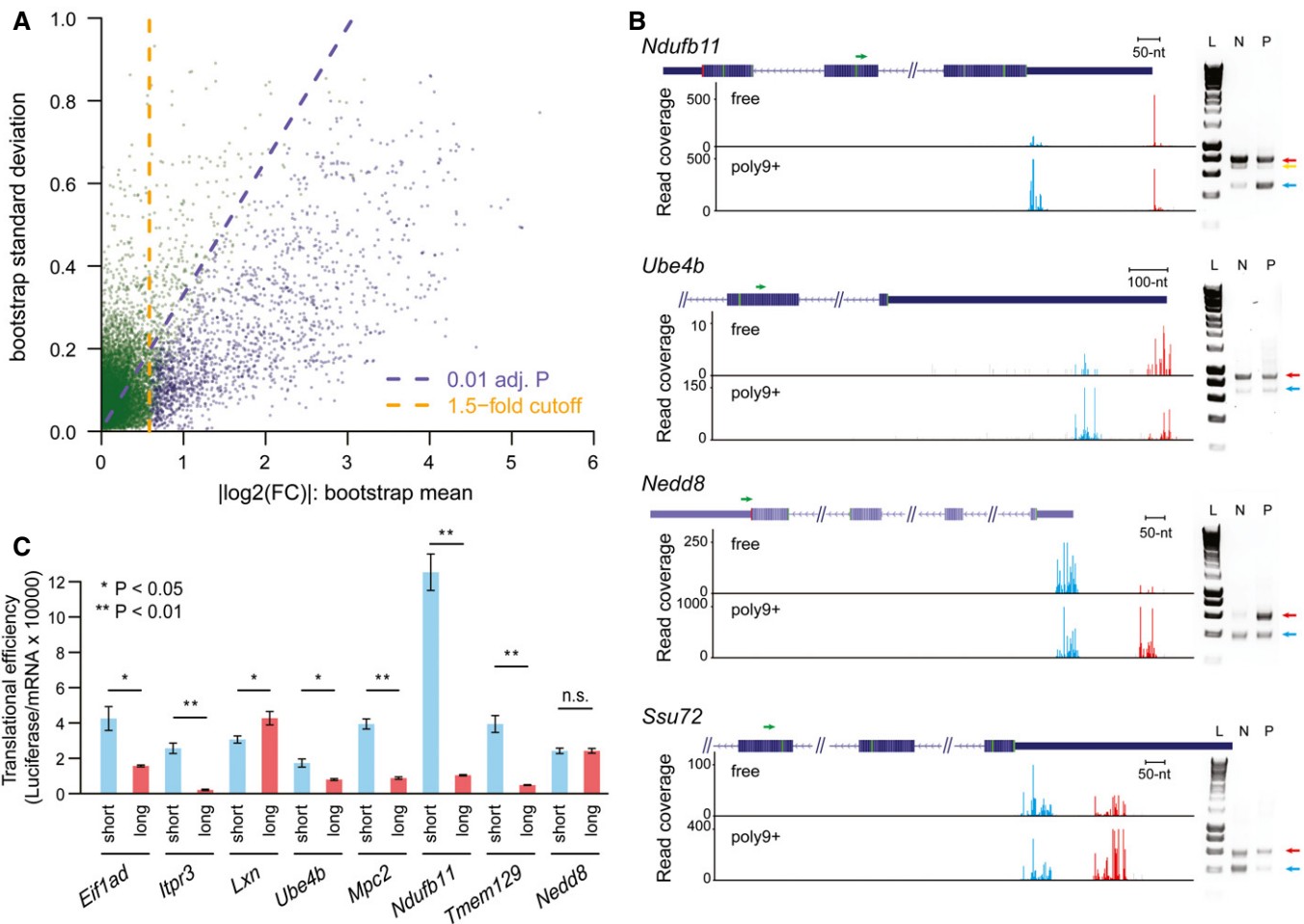

**Figure 2.  Alternative TSSs lead to significantly differential TE in 745 out of 4,153 multi-TSS genes.**

A   Scatter plot showing the bootstrap means (*x*-axis) and standard deviations (*y*-axis) for log2-transformed TE difference between 13,118 TSS isoform pairs in the 4,153 multi-TSS genes. Dashed purple lines indicated the Benjamini–Hochberg adjusted *P*-value of 0.01, and dashed orange lines indicated the 1.5-fold divergence. Genes with significant TE divergence (Benjamini–Hochberg adjusted *P*-value < 0.01, TE divergence > 1.5-fold) are depicted in blue. See also Table EV2.

B   Independent validation of TSS isoforms and their associated translational efficiency in genes *Ndufb11*, *Ube4b*, *Nedd8*, and *Ssu72*, respectively. Left: Under each gene structure, cumulative reads were shown for the alternative TSSs in the "free" fraction and poly9+ fraction. Green arrows above the gene structure indicate the locations of the reverse PCR primer. Red and blue bars represented sequencing reads mapped within distal and proximal TSSs, respectively; gray bars represented reads mapped outside of the identified TSSs. Right: Agarose gel electrophoresis of amplified products of mRNA 5′ ends obtained from non-ribosomal fraction and polysomal fraction. Positions of the distal TSS isoform and the proximal TSS isoforms are indicated with red and blue arrows, respectively. In the case of gene *Ndufb11*, the band below the distal TSS (indicated by a yellow arrow) in the gel image was caused by an alternative splicing event, which removed an 88-nt region for a minor fraction of transcripts initiating at the distal TSS. L, HyperLadder I; N, non-ribosomal fraction; P, polysomal fraction. The description of these genes can be found in Table EV3.

C   Alternative 5′UTR sequences are able to drive the observed isoform-specific TE divergence. An *in vivo* reporter system was used to compare the TE of a *Renilla* luminescent reporter gene led by the 5′UTR sequences derived from eight pairs of alternative TSS isoforms identified in eight genes. TE is calculated by luciferase activity normalized to mRNA abundance. Seven out of eight reporter pairs showed significant differential TE biased toward the same TSS isoforms as observed in our global analysis (*n* = 3; mean ± SEM; \**P* < 0.05, \*\**P* < 0.01; Student's *t*-test). The description of these genes can be found in Table EV3.

uORFs between the alternative isoforms contributed to the observed TE difference, we first separated the isoform pairs into two groups according to the presence of uORFs in the divergent 5′UTRs. Comparing the distribution of isoform-specific TE divergence between the two groups, we observed significant differences such that the presence of uORFs led to larger TE decrease in the longer isoforms (Fig 4A). Indeed, for the isoform pairs with longer one translating less efficiently, uORFs appeared in their divergent 5′UTRs at a significantly higher frequency than for the remaining

pairs (85.5% versus 30.2%, *P* = 1.0e-44, Fisher's exact test). We then further checked whether the number of uORFs was correlated with the degree of translation inhibition. Consistent with previous report (Calvo, 2009), as shown in Fig 4A, more uORFs in the divergent 5′UTRs resulted in larger TE decrease in the longer isoforms.

In our previous study (Hou *et al*, 2015), we observed that out-of-frame and in-frame uAUGs conferred different effects on translational regulation—whereas out-of-frame uAUGs tended to decrease TE, in-frame uAUGs did not show significant impact. Using the

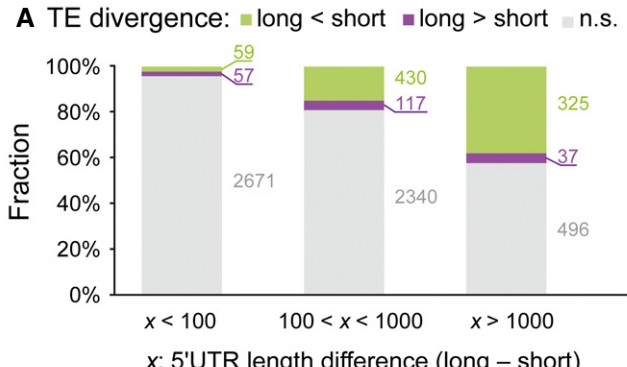

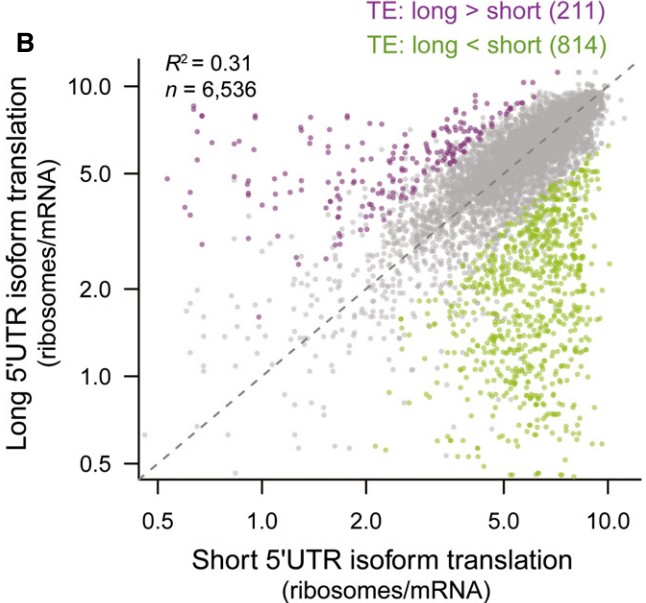

**Figure 3. Isoforms with longer 5′UTR tend to have lower TE.**

A Barplots showing the fraction of alternative TSS isoform pairs with and without significant differential TE. Isoform pairs with certain 5′UTR length difference were grouped together. The larger the length difference between the two isoforms, the higher the fraction associated with significant TE divergence.

B Scatter plot comparing the number of ribosomes per mRNA between shorter 5′UTR isoforms (*x*-axis) and longer 5′UTR isoforms (*y*-axis) from the same genes. Purple and green dots were isoform pairs with significant differential TE biased toward longer and shorter isoforms, respectively.

same analysis as for uORFs, here we also checked for the two subtypes of uAUGs separately. In consistence with our previous findings, the presence of out-of-frame uAUGs but not the in-frame ones in the divergent 5′UTRs led to the decreased TE of the longer isoforms (Fig 4B and C).

The analysis of uORFs/uAUGs described above was based on the presence of canonical start codon (AUG) in the 5′UTR sequences. However, some of these uORFs/uAUGs might not be used in the 3T3 cells, and it has been shown that mRNA translation, particularly for uORFs, could initiate from non-canonical start codons (Ingolia *et al*, 2011; Fritsch *et al*, 2012). To further substantiate the observed

negative impact of upstream start codons on translation of main ORFs, we ascertained a set of uORFs/uAUGs supported with experimental evidence. Based on initiating ribosome profiling we performed for this purpose, together with published 3T3 ribosome footprinting data (Shalgi *et al*, 2013), we identified a total of 163 canonical uORFs and 9 out-of-frame uAUGs using the ORF-RATER tool developed recently (Fields *et al*, 2015). Restricting the above analyses to these uORFs/uAUGs, we again witnessed the same regulatory tendency (Fig 4D and E). Intriguingly, such effects were neither observed for uORFs led by non-canonical start codons (CUG, GUG, or UUG), nor for out-of-frame non-canonical upstream start codons (Figs 4F and G, and EV5A and B).

**5′ cap-adjacent stable RNA secondary structures inhibit translation**

*In vitro* experiments have shown stable RNA secondary structures in vicinity of mRNA 5′ ends could diminish translation initiation (Kozak, 1989). To check whether such observation also holds true *in vivo* and whether the presence/absence of RNA secondary structures close to 5′ ends could contribute to the observed TE difference between alternative TSS isoforms, we calculated and compared the minimum free energy (MFE) in the regions immediately following the alternative TSSs. Compared to isoform pairs that had stable structures immediately after TSSs (MFE < −30 kcal/mol for 50-nt RNA fragments) in both or neither of the isoforms, the genes with stable RNA structures only in one isoform showed significantly different TE divergence between the two isoforms. Apparently, the presence of stable RNA structures near 5′ cap led to translational repression (Fig 5A; see Fig EV6A and B for two examples), indicating such negative impact on translation observed previously *in vitro* also worked *in vivo* as a general regulatory mechanism. Beyond the region immediately after TSSs, as shown in Fig 5B, stable RNA structures (MFE < −35 kcal/mol in 50-nt RNA fragments) still conferred negative impact on translation, although much weaker. The results remained the same if ensemble free energy (EFE) instead of MFE was used (Fig EV6C and D).

**TSS isoforms with 5′ TOP sequences are translated less efficiently**

Another category of known translational regulatory elements located close to TSSs is 5′ TOP, which is a highly conserved sequence stretch consisting of a C residue at the cap site, followed by 4–14 pyrimidines (Meyuhas *et al*, 1996). 5′ TOP is a sequence hallmark for most vertebrate mRNAs that encode ribosomal proteins and translation elongation factors (Meyuhas, 2000). Protein synthesis of these TOP genes is highly sensitive to cell growth rate, with growth arrest leading to strong inhibition of their translation (Meyuhas, 2000). To check whether the presence/absence of 5′ TOP sequence contributed to the observed TE divergence between TSS isoforms, we collected 166 known TOP genes from literature (Hsieh *et al*, 2012; Thoreen *et al*, 2012). Among these, 33 genes expressed multiple TSS isoforms in 3T3 cells, of which one isoform contained 5′ TOP sequences (C followed by at least 4 pyrimidines). Comparing to the isoforms from the same gene but without 5′ TOP sequences, the TOP-containing isoforms tended to translate significantly less efficiently (Fig 5C). Given that our study was performed in cells under normal growth condition, this observation suggests the TOP

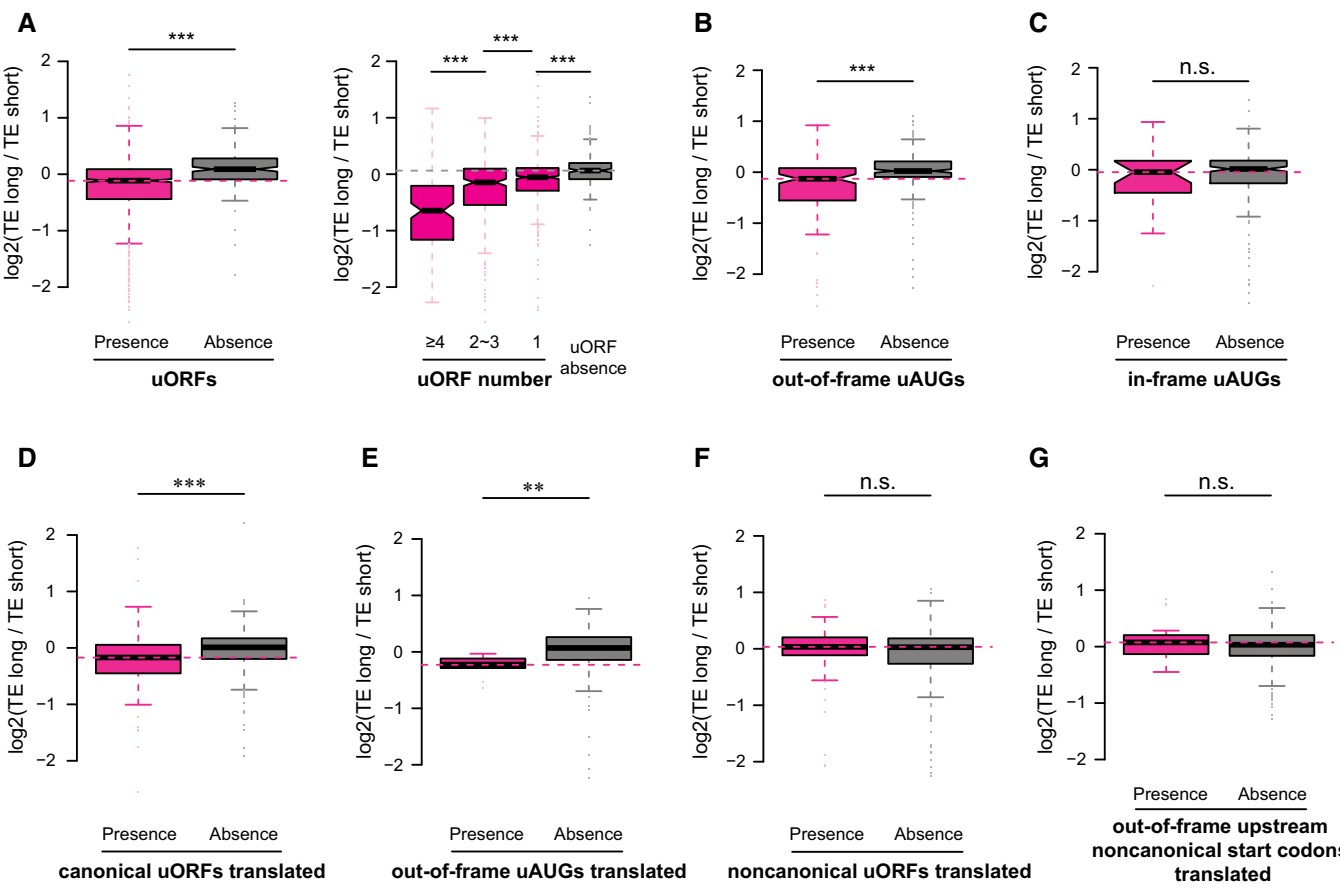

**Figure 4.   Upstream translation started at AUG negatively affects the main ORF translation.**

A   Left: Boxplots comparing the log2 TE fold changes between two groups of alternative isoform pairs, one group with at least one uORF present in the isoform-divergent 5′UTR and the other without. Right: The group with uORF was further separated into three subgroups according to the number of uORFs present in the divergent 5′UTR.
B   Same as (A)—left, but the sequence feature of interest is the out-of-frame uAUGs.
C   Same as (A)—left, but the sequence feature of interest is the in-frame uAUGs.
D   Same as (A)—left, but the sequence feature of interest is the translated uORFs (i.e. supported by ribosome footprinting) with canonical AUG start codon.
E   Same as (A)—left, but the sequence feature of interest is the translated out-of-frame uAUGs (i.e. supported by ribosome footprinting).
F   Same as (A)—left, but the sequence feature of interest is the translated uORFs (i.e. supported by ribosome footprinting) with non-canonical start codons.
G   Same as (A)—left, but the sequence feature of interest is the translated out-of-frame upstream non-canonical start codons (i.e. supported by ribosome footprinting).

Data information: **$P < 0.01$, ***$P < 0.001$; Mann–Whitney $U$-test. Box edges represent quantiles, whiskers represent extreme data points no more than 1.5 times the interquartile range.

sequences may to some extent repress translation even without growth arrest.

## Novel sequence motifs associated with isoform-specific translation

To further extract potential regulatory sequence elements, we extended our sequence feature analyses by correlating the appearance of all hexamers in the divergent 5′UTRs to the observed TE difference. As AUG-containing hexamers may reflect the presence of uORFs or uAUGs, they were excluded for this analysis. In total, we identified 137 hexamers significantly correlated with the observed TE divergence (BH-corrected $P$-value < 0.01), all of which acted negatively on translational regulation (Table EV4). For instance, the presence of hexamer AAAAAU, which matches the binding motif of PABPC1 (Paz *et al*, 2014), attenuated TE significantly (adjusted

$P = 6.4e\text{-}04$). Interestingly, although PABPC1, a cytoplasmic poly(A) binding protein, typically binds to 3′ poly(A) tails of eukaryotic mRNAs, it has been shown that PABPC1 binding to an A-rich elements in its own 5′UTR could inhibit its translation (de Melo Neto *et al*, 1995; Melo *et al*, 2003). To substantiate our findings on other hexamer motifs, we chose two hexamers (AAUCCC and CAAGAU) for validation using reporter assays (Materials and Methods). As illustrated in Fig 5D and E, the presence of five copies of each of the two motifs in 5′UTRs indeed decreased the translation of the luciferase reporter gene.

## Quantitative models explaining the TE difference between alternative TSS isoforms

The analyses so far have revealed a variety of sequence features mediating TE regulation between alternative TSS isoforms. To

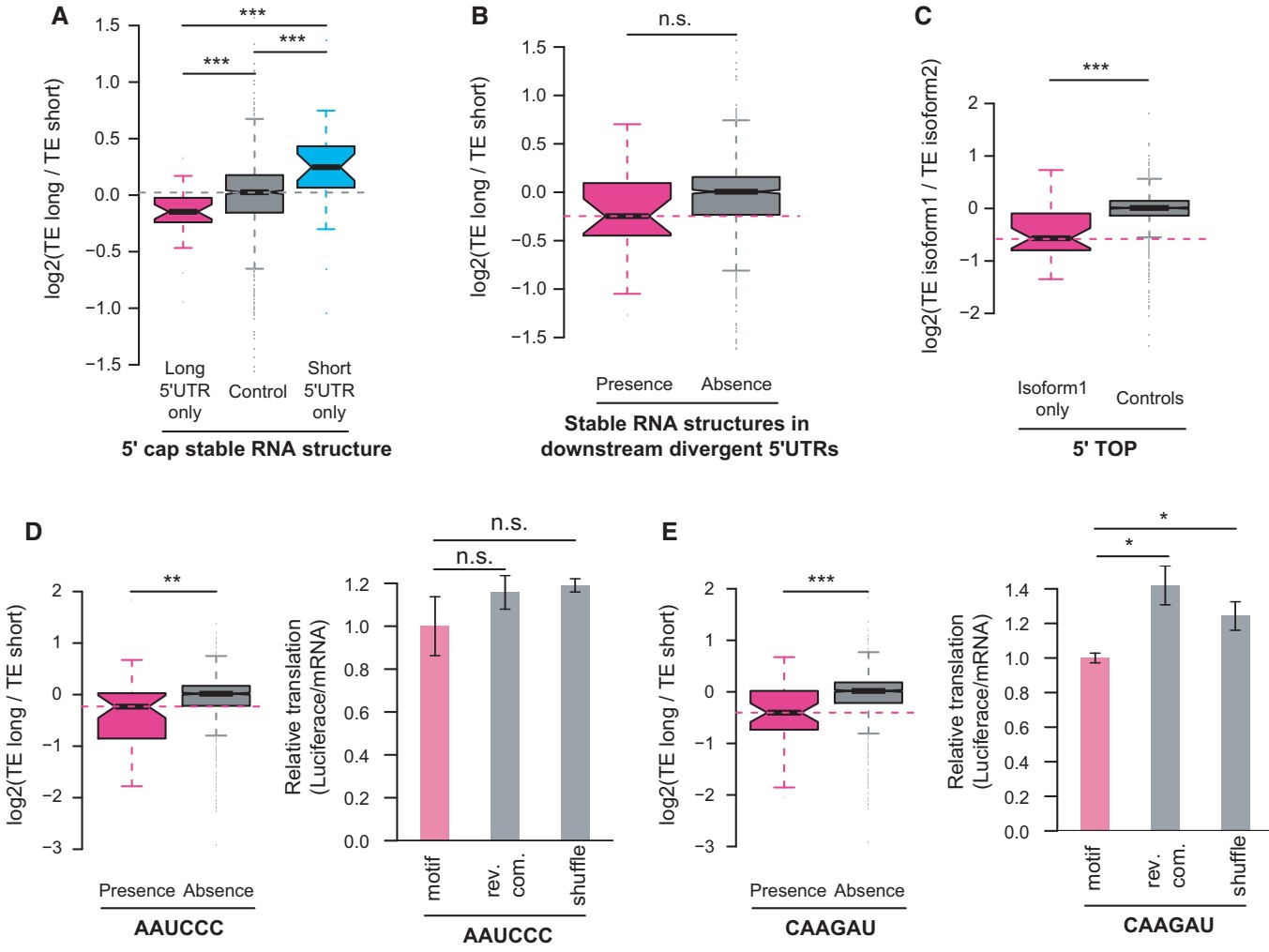

**Figure 5.  Roles of stable RNA structures, 5′ TOP sequences, and sequence motifs within 5′UTR for translational regulation.**

A    Boxplots comparing the log2 TE fold changes between three groups of alternative isoform pairs, the first group with 5′ cap-adjacent (50 nt to 5′ ends) stable RNA secondary structures (MFE < −30 kcal/mol) present only in long 5′UTR isoforms, the second group with 5′ cap-adjacent stable RNA structure present/absent in both isoforms, and the last group with 5′ cap-adjacent stable RNA structure present only in short 5′UTR isoforms.

B    Boxplots comparing the log2 TE fold changes between two groups of alternative isoform pairs, one group with stable RNA secondary structures (MFE < −35 kcal/mol in any 50-nt RNA fragments) present in the downstream divergent 5′UTR and the other without.

C    Boxplots comparing the log2 TE fold changes between TOP genes and non-TOP genes (controls). For TOP genes, the TE fold changes were the ratios between the isoforms with 5′ TOP sequences present and isoforms without, and for non-TOP genes, isoforms were randomly assigned as numerators and denominators.

D    Left: Boxplots comparing the log2 TE fold changes between two groups of alternative isoform pairs, one group with the motif AAUCCC present in divergent 5′UTRs and the other without.
      Right: Luciferase assay comparing the relative TE between reporter genes with five copies of motif AAUCCC, reverse complement of motif AAUCCC, and randomly shuffled sequences in their 5′UTRs (*n* = 3; mean ± SEM; n.s. *P* > 0.05).

E    Similar to (D), but the motif is CAAGAU (*n* = 3; mean ± SEM; *\*P* < 0.05; Student's *t*-test).

Data information: In boxplots, *\*P* < 0.05, *\*\*P* < 0.01, *\*\*\*P* < 0.001; Mann–Whitney *U*-test. Box edges represent quantiles, whiskers represent extreme data points no more than 1.5 times the interquartile range.

further understand the relative contribution of these elements to the observed TE divergence, alone or in combination, we trained nonlinear regression models (Materials and Methods). As shown in Fig 6A, as individual features, the number of uORFs in the divergent 5′UTRs and the 5′UTR length difference between the two isoforms were the two best single predictors for TE difference, which explained 35.5 and 35.1% of its variance, respectively. The number of out-of-frame AUGs and the appearance of stable RNA secondary structures near 5′ ends had less prediction

power, probably due to their limited occurrence in our dataset, yet explaining the difference by 3.7 and 3.5%, respectively (Fig 6A). In combination, the model integrating all the features explained 57% of the variance of observed TE difference (Figs 6B and EV7). To further test the predictive power in model generalization, 10-fold cross-validation procedure was applied, in which models trained on nine tenth of all isoform pairs with significant TE divergence were used to predict the observed TE difference for the remaining one tenth pairs. In 100 times randomly partitioning

**A**   Contribution of sequence features in the models

| Sequence Feature | Individual | Cumulative | Delta cumulative |
|---|---|---|---|
| uORF | 35.6% | 35.6% | 35.6% |
| divergent 5′UTR length | 35.1% | 41.3% | 5.8% |
| out-of-frame uAUG | 3.7% | 43.8% | 2.4% |
| 5′ cap RNA structure | 3.5% | 43.9% | 0.1% |
| downstream RNA structure | 2.4% | 45.7% | 1.8% |
| 5′ TOP sequence | 1.7% | 46.5% | 0.7% |
| Hexamers | * 23.2% | 56.8% | 10.3% |

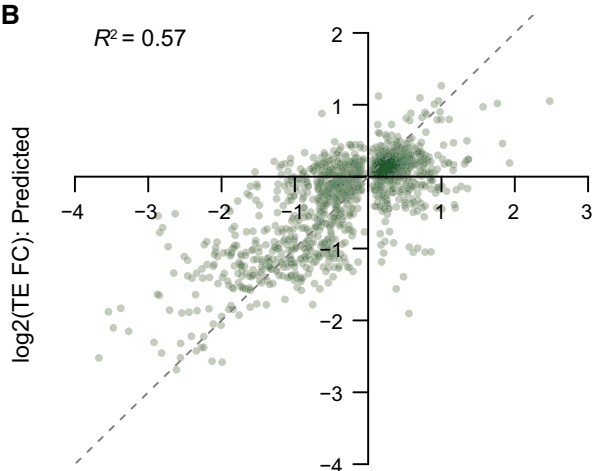

**B**   $R^2 = 0.57$

log2(TE FC): Predicted

log2(TE FC): Observed

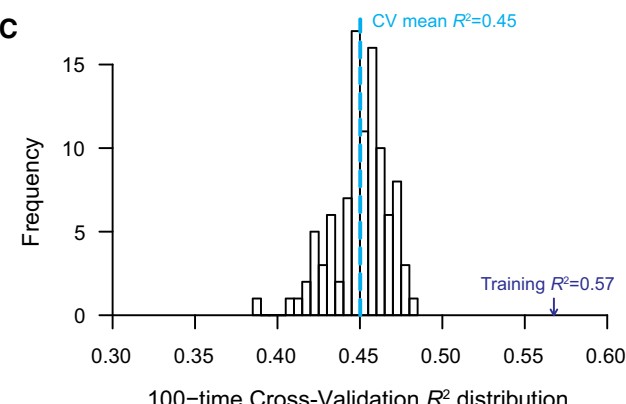

**C**

CV mean $R^2$=0.45

Training $R^2$=0.57

Frequency

100−time Cross-Validation $R^2$ distribution

---

**Figure 6.  Quantitative model explaining the TE difference between alternative TSS isoforms.**

A   Barplots showing the individual and cumulative contribution for sequence features in explaining the TE difference between alternative TSS isoforms. Individual: variance of TE divergence explained by the model with only the sequence feature; Cumulative: variance explained by the model combining the sequence feature and those above; Delta cumulative: additional variance explained by adding the sequence feature to the model. *The value was the contribution of all significant hexamer motifs.

B   The combinatory nonlinear regression model based on all sequences features investigated in this study explained 57% variance of TE difference between alternative TSS isoforms.

C   Histogram showing the distribution of model-explained variance in the 100 times cross-validation procedure.

---

of training and test datasets, the models on average explained 45 % of the variance of TE difference observed between the test isoform pairs (Fig 6C).

## Discussion

In the multi-step process of eukaryotic protein biogenesis, transcription initiation serves as the first layer in the control of gene expression. Transcription initiated from alternative promoters usually leads to the formation of mRNA transcripts sharing the same coding sequences yet different 5′UTRs, thereby subject to potential differential translational regulation. Although the functional significance of such coordination between transcription and translation through "writing" and "reading" alternative 5′UTRs has been demonstrated for a handful of genes, the prevalence of the 5′UTR-isoform-specific translational control across a mammalian genome is currently unknown. Here, we for the first time report a genome-wide survey of the interdependence between transcription and translation in mammalian cells by combining polysome profiling and mRNA 5′ end sequencing. Our data revealed substantial coordinated regulation of the two processes via alternative TSS usage: around half of expressed genes initiated their transcription at multiple sites, nearly 20% of which showed significant translational difference between alternative TSS isoforms. The large set of genes with TSS isoform-divergent translational regulation collected in this study also enabled systematical characterization of the regulatory effect of diverse sequence features embedded in 5′UTRs.

Three lines of evidence demonstrated that our approach faithfully measured the translational status associated with distinct TSS isoforms. First, at the gene level, TE values estimated based on our polysome profiling correlated well with those based on both ribosome footprinting and mass spectrometry-based proteomics measurement. Second, for four randomly chosen multi-TSS genes, we validated the identified TSS isoforms and their differential translational status using an independent experimental strategy. Finally, using an *in vivo* reporter assay, we demonstrated that alternative 5′UTR sequences could drive the TE divergence observed in our global analysis.

A recent study combining polysome profiling with RNA-seq (TrIP-seq) sought to address the mRNA isoform-specific translational control in a comprehensive manner (Floor & Doudna, 2016). One of their observations that the predominant contribution to isoform-specific translational status came from sequence features in 3′UTRs over those in 5′UTRs, agreed neither with our results, nor with a previous direct survey on the translational impact of 3′UTR diversity (Spies *et al*, 2013). Indeed, the 3′UTR study, also performed in 3T3 cells, found that alternative 3′UTRs had only modest effect on TE, suggesting 3′UTR isoform choice plays a minor role in regulating translation. Although the inconsistent observations between these studies could be attributed to the large difference in the cell types studied (mouse NIH3T3 versus human HEK 293T), a more possible explanation lies in the different strategies used to quantify isoform abundance. Whereas both studies in 3T3 cells applied targeted experimental approaches to directly measure the expression of isoforms with distinct 5′ (5′ end sequencing in this study) or 3′ ends (3P-/2P-seq in Spies *et al*, 2013), TrIP-seq used Cufflinks suite to estimate the abundance of isoforms resulted from both 5′ and 3′ end diversity as well as alternative splicing. On one hand, whereas the latter study could offer better insights into the relative contribution of different regions along the transcripts, the other two focused on one specific UTR and might both over- and underestimate the contribution from their targeted regions. For

instance, if a pair of alternative 5′ end and 3′ end concurred in one transcript, the observed translational status associated with that specific transcript might be erroneously attributed to either region under study, thus generating false-positive findings. In contrast, the effect from one region could be offset by the opposite impact from the other region, thus resulting in false negatives. While such scenarios may exist and can even explain some of our observed isoform-TE differences that could not be fully accounted by the features investigated in this study, we believe they do not affect our general conclusions. Based on the published 3P-/2P-seq data (Spies et al, 2013), we separated the 4,153 multi-TSS genes into two groups: 1,841 with one 3′ end and 1,767 with multi-3′ end. For both groups, a similar percentage (353/1,841 versus 289/1,767) showed significant TE divergence between alternative TSS isoforms. The results from our sequence feature analyses also held true even if restricted to either gene groups (Fig EV8). On the other hand, more importantly, compared to isoform abundance estimation based solely on RNA-seq, direct isoform profiling using targeted approaches undoubtedly provides more accurate quantification. Particularly, about half of the TSSs identified in this study were not annotated in either RefSeq or Ensembl. Those unannotated isoforms would be overlooked in the TrIP-seq analysis based on available annotation. In addition, our conclusions on the substantial impact of 5′UTRs are supported by our previous observation that the SNPs responsible for allele-specific TE were enriched in 5′UTRs (Hou et al, 2015). Mechanistically, the stronger regulatory impact of 5′UTRs on translation in general agrees with the notion that translation initiation is the rate-limiting step with higher regulatory potential (Sonenberg & Hinnebusch, 2009).

Among the isoform pairs showing significant TE difference, longer isoforms were in general associated with lower TE; such trend became more apparent with increased length difference between isoform pairs, suggesting that sequence features embedded in 5′UTRs acted more frequently to repress than to enhance translation. Consistent with this, all the regulatory features that we identified by comparing TE between isoform pairs were repressive elements. Notably, our study was performed in fast-growing fibroblasts; whether this result could be generalized awaits future work on translational regulation in various cell types and/or under diverse conditions.

Previous genome-wide studies investigating cis-regulatory elements in translational control have been mainly based on comparisons across different genes comprised of diverse CDS and UTRs, in which complex regulatory effects could not always be easily disentangled (Brockmann et al, 2007; Wu et al, 2008; Vogel et al, 2010). In contrast, our study focused on the TE difference between alternative TSS isoforms derived from the same gene, most of which shared the same CDS, and even 3′UTRs. Therefore, the confounders from outside of 5′UTRs were largely excluded, to achieve both improved sensitivity and specificity in detecting regulatory elements in 5′UTRs. As a result, our nonlinear regression model integrating all the features identified in this study explained over half of the variance of the observed TE difference between isoforms. While some of the remaining unexplained effects could still come from 5′UTR-coupled alternative CDS and/or 3′UTRs, we believe the majority may result from other unanalyzed features in 5′UTRs, such as RNA modification (e.g. m$^6$A), internal ribosome entry sites (IRES), and SINEUP binding sites (Carrieri et al, 2012; Meyer et al,

2015; Zhou et al, 2015; Zucchelli et al, 2015; Weingarten-Gabbay et al, 2016).

Based on luciferase reporter assays performed across cell lines of various tissue origins, the TrIP-seq study revealed that while 3′UTRs tend to confer cell type-specific translational regulation, 5′UTRs seem to exert coherent regulation between cell lines (Floor & Doudna, 2016). This is consistent with our observation that the majority of the features identified in 5′UTRs mediate translational regulation through interfering with basic translational machinery. Beyond previously reported functional consequence of these features, our results still offer novel insights. For example, we have demonstrated the strong negative effect of uORFs and showed that their occurrence in the divergent 5′UTRs was the single best predictor for TE difference between alternative TSS isoforms. Interestingly, we found that only the uORFs with canonical AUG start codon could exert such negative regulation. Even though the non-cognate start codons, in particular CUG, composed more upstream translation initiation sites based on ribosome footprinting data, they showed no significant effects. A recent genome-wide study of translation initiation sites also found that uORFs led by non-optimal AUG variants were translated in parallel to the downstream main ORF, whereas uORFs starting with AUGs in an optimal context often repressed the main ORF translation (Lee et al, 2012). Both studies support the leaky scanning theory and suggest that the accessibility of main ORF start codons to the initiation complex depends on the context of upstream start codons (Michel et al, 2014). Intriguingly, compared to non-cognate start codons, AUG is highly depleted in the 5′UTR sequences from the mouse genome and many other species (Churbanov et al, 2005; Iacono et al, 2005; Neafsey & Galagan, 2007), indicating that the promiscuous presence of uORFs with strong regulatory impact is under purifying selection. An earlier in vitro experiment reported two types of RNA secondary structures that could inhibit translation in cis: One stem-loop structure positioned immediately after 5′ cap prevented mRNA from engaging 40S subunits, and the other more stable stem-loop positioned further downstream stalled the scanning 40S subunits (Kozak, 1989). In this study, we recapitulated these two phenomena and provided the first genome-wide in vivo evidence for the hypothesized mechanisms for stable mRNA structures in 5′UTRs to reduce TE. 5′ TOP has been known to repress the translation upon cell growth arrest. Our study performed in cells under normal growth condition suggested that TOP sequences could also exert the repressive effect without growth arrest, probably to a much lesser extent.

This study has been mainly focused on the impact of alternative TSSs on quantitative changes of TE, therefore we restricted our analyses to the alternative TSSs altering only 5′UTRs. Besides, alternative TSSs can also lead to transcripts with different ORFs and expand the repertoire of encoded proteins by, for example, diversifying protein N-termini (Pelechano et al, 2013), which is often essential for proper protein functions and/or their subcellular localization (Chen et al, 2002; Arce et al, 2006; Zhang et al, 2015). Very recently, a novel isoform of anaplastic lymphoma kinase (ALK) gene was reported in human carcinoma, which initiated at a cryptic TSS located in intron 19. This novel isoform can produce N-terminal truncated proteins that could promote tumorigenesis by stimulating multiple oncogenic signaling pathways (Wiesner et al, 2015). In 3T3 cells, among the 5,324 TSSs located downstream of annotated start codons, 502 expressed at a decent level and were associated with

heavy polysome ($\geq 4$ ribosomes/mRNA), of which 71 contained downstream translation initiation sites supported by the ribosome footprinting data (Table EV5, Fig EV9A). Collectively, these observations indicate that the transcripts led by downstream TSSs could be actively translated, yielding N-terminal truncated proteins. Similarly, we also identified several instances where alternative TSSs can lead to the transcripts encoding N-terminal extended proteins (Fig EV9B). Future work would be needed to decipher the functions and regulatory mechanisms of these novel protein isoforms.

Finally, our study revealed substantial interdependence between transcription initiation and translational regulation in one cell type under normal growth condition. Future application of our approach in multiple tissues and under different conditions will facilitate the elucidation of tissue- and condition-specific regulation, which could in turn unveil the role of such coordinated regulation during development as well as in human diseases.

# Materials and Methods

### Cell culture

Mouse NIH3T3 cells were used and cultivated in Dulbecco's modified Eagle's medium (DMEM) supplemented with 10% fetal bovine serum (FBS, Gibco) at 37°C with 5% $CO_2$ and split every second or third day.

### RNA sequencing

Total RNAs from mouse NIH3T3 cells were extracted using TRIzol reagent (Life Technologies) following the manufacturer's protocol. TruSeq Stranded Total RNA library was prepared with 500 ng total RNA according to the manufacturer's protocol (Illumina). The libraries were sequenced in 1× 100 nt manner on HiSeq 2000 platform (Illumina).

### Polysome profiling

Mouse NIH3T3 cells were grown to 80% confluency. Prior to lysis, cells were treated with cycloheximide (100 μg/ml) for 10 min at 37°C. Then, cells were washed with ice-cold PBS (supplemented with 100 μg/ml cycloheximide) and further lysed in 300 μl of lysis buffer (10 mM HEPES pH 7.4, 150 mM KCl, 10 mM $MgCl_2$, 1% NP-40, 0.5 mM DTT, 100 μg/ml cycloheximide). After lysing the cells by passing eight times through 26-gauge needle, the nuclei and the membrane debris were removed by centrifugation (16,000 $g$, 10 min, at 4°C). The supernatant was layered onto a 10 ml linear sucrose gradient (10–50% [w/v]), supplemented with 10 mM HEPES pH 7.4, 150 mM KCl, 10 mM $MgCl_2$, 0.5 mM DTT, 100 μg/ml cycloheximide), and centrifuged in a SW41Ti rotor (Beckman) for 120 min at 160,000 $g$ at 4°C. Fractions were manually collected according to the A254 peaks that indicate the number of ribosomes. 50 ng fly total RNAs were added into each fraction as spike-in immediately. The collected fractions were then digested with 200 μg proteinase K in 1% SDS for 30 min at 42°C. RNA from each fraction was recovered by extraction with an equal volume of acid phenol–chloroform (pH 4.5), followed by ethanol precipitation.

### 5′ end sequencing

Three microgram total RNAs collected from each fraction (see above) were reverse-transcribed using random primer (N15-oligo) tailed with 3′ part of Illumina TruSeq Universal Adaptor sequence (P5). 5′ complete single-stranded cDNAs were captured based on a protocol from Takahashi *et al* (2012) with minor modification. In brief, cap structure and 3′ ends of all RNAs were oxidized by $NaIO_4$ on ice for 45 min, followed by an overnight biotinylation with a long-arm biotin hydrazide at room temperature. Single-stranded RNA regions that were not covered by synthesized cDNAs including the 3′ ends were cleaved using RNase I. The 5′ complete cDNAs containing the biotinylated cap site were then captured with Dynabeads® M-280 Streptavidin (Life Technologies). RNAs were hydrolyzed with 50 mM NaOH and single-stranded cDNAs were released from the beads. After ligation with double-stranded 5′ linkers with random overhangs (containing 3′ part of Illumina TruSeq Universal Adaptor P7), cDNAs were amplified for 18 cycles using cap forward primer (containing P5) and cap reverse primer with barcode included. The amplified libraries were sequenced in $2 \times 100$ nt manner on Illumina HiSeq2000 platform. All the primer and adaptor sequences were listed in Table EV6.

### 5′ end sequencing read processing and TSS cluster identification

The paired-end reads were first subjected to adapter removal using FLEXBAR with the following parameters: -u 2 -m 48 -ae RIGHT -at 2 -ao 1 (Dodt *et al*, 2012). Then, the first 15 nt of the 1st read derived from the random primer region was further removed due to potential high mismatches. Read pairs that were concordantly mapped to the reference sequences of rRNA, tRNA, snRNA, snoRNA, and miscRNAs (available from Ensembl and RepeatMasker annotation) using Bowtie 2 (version 2.1.0) (Langmead & Salzberg, 2012) with default parameters (in --end-to-end & --sensitive mode) were excluded. The remaining reads were then mapped to the mouse reference genome (mm10, downloaded from UCSC Genome Browser) using Tophat2 (version 2.0.10) (Kim *et al*, 2013) with the parameters --mate-inner-dist 200 --mate-std-dev 100 -N 3 --read-gap-length 2 --read-edit-dist 3 --min-anchor 6 --library-type fr-firststrand --segment-mismatches 2 --segment-length 25 with the input of Ensembl mouse gene annotation (Release 72). Reads that were mapped to multiple genomic loci and the two reads in one pair that were mapped to different chromosomes were discarded in following analysis. Compared to the RIKEN CAGE protocol that produces 27-nt reads (Takahashi *et al*, 2012), our 5′ end sequencing approach yields much longer reads, which significantly increases the percentage of uniquely mapped reads.

For each of the uniquely, concordantly mapped read pairs, only the 5′ end position of its 2nd read (termed as tags hereafter), which corresponds to the 5′ end of RNA transcripts, was used for determining TSS cluster. To increase the sensitivity in detecting the expressed TSSs, we combined the sequencing data from the seven fractions in each replicate together. Briefly, genomic positions with tags beyond local background and within a distance of 20 nt were assigned into one cluster. Here, the local background (bg) for each position was determined by the maximum of (i) local expectation, that is the average tag coverage in the window of 500 nt centered at the position, and (ii) expression background, that is the sequencing

depth-normalized RNA-seq read coverage within the window from 500 nt upstream to 1,500 nt downstream of the position. In order to improve the spatial resolution in detecting TSSs, for the clusters longer than 100nt, we stepwise increased the local background by 0.5 bg up to 3 bg, until all the sub-clusters with tag coverage beyond the increased background are shorter than 100 nt. To further decrease the potential false-positive findings, all clusters from the two replicates were subjected to irreproducible discovery rate (IDR) analysis using the IDR Python package (https://github.com/nboley/idr, version 2.0.1) with parameters "--input-file-type narrowPeak --rank signal.value", where signal.value was the tag counts in each cluster. TSS clusters identified in both replicates with IDR $\leq 0.05$ were kept, and on average 85% of all tags were located within these TSS clusters, indicating the high quality of our 5′ end sequencing data. These TSS clusters were then assigned to protein-coding genes based on RefSeq gene annotation. Here, we only retained the TSS clusters in the gross 5′UTRs for downstream analysis. The gross 5′UTR included annotated 5′UTRs (from the most 5′ annotated TSS to the most 3′ annotated start codon) and 1 kb upstream of the most 5′ annotated TSSs.

## Across-fraction data normalization using *D. melanogaster* spike-in RNA

After trimming and filtering (see above), 5′ end sequencing reads were simultaneously mapped to the *D. melanogaster* reference genome (dm3, downloaded from UCSC Genome Browser) using Tophat2 (version 2.0.10) with the same parameters as described above, and with the input of RefSeq fly gene annotation (downloaded from UCSC Genome Browser). TSS clusters in the *D. melanogaster* genome were identified as described above. TSS clusters with more than 10 tags in each of the seven fractions in both replicates were kept, and the upper quantile of these tag counts in each fraction was taken as the normalization factors to normalize the tag counts of mouse TSS clusters from the corresponding fraction.

## TSS isoform-specific translational efficiency (TE) calculation

Given the normalized tag count $C_{ij}$ for TSS isoform $i$ in fraction $j$, we determined the total amount of the isoform as $T_i = \sum_j C_{ij}$. According to the profile of the sucrose gradient, we calculated the overall number of ribosomes associated with the TSS isoform $i$ as $R_i = \sum_j r_j C_{ij}$ where $r_j$ is the average ribosome number in the $j$th fraction (fractions corresponding to free RNP and 40S/60S $r_1 = r_2 = 0$, 80S monosome fraction $r_3 = 1$, and polysome fractions $r_4 = 2.5$, $r_5 = 4.5$, $r_6 = 7.5$, and $r_7 = 12$). The translational efficiency was then calculated as average number of ribosomes associated with each TSS isoform in unit ORF length, that is, $TE_i = R_i/l_i/T_i$, where $l_i$ is the length of the corresponding ORF.

## Determination of TE divergence between TSS isoforms

For each of the multi-TSS genes, we performed pairwise comparison of the TE associated with different TSS isoforms. To account for the uncertainty in estimating TE divergence between two isoforms, we performed a bootstrapping-based test to assess statistical significance. In brief, from the 5′ end sequencing data of each fraction, we generated pseudo-datasets of the same depth by sampling all uniquely mapped reads at random with replacement. After recounting tags in each TSS cluster from each fraction in the pseudo-dataset, we recomputed the average ribosomes per mRNA associated with each TSS and then the log2-transformed TE fold changes between every TSS isoform pair. After repeating the bootstrapping procedure 1,000 times, we obtained for each pairwise comparison a distribution of log2-transformed TE fold changes, which were then summarized into a mean and a standard deviation. The bootstrapping means correlate perfectly with the TE fold changes calculated in the real data ($r = 0.9997$). Nonzero bootstrapping means indicate that the two TSS isoforms are translated with different efficiency. To determine the statistical significance of such difference, we calculated a $P$-value based on the $Z$-score that represented how many folds of standard deviation the bootstrapping mean deviated from zero. The raw $P$-values were then adjusted using the Benjamini–Hochberg method. To determine the false discovery rate (FDR), we applied a similar label permutation strategy as used previously (Hou *et al*, 2015). In short, pairwise comparison labels were shuffled for 100 times in both replicates, and in each of the 100 shuffled sets, we counted the number of comparisons in both replicates meeting the fold change (FC) requirement ($|FC| > x$) and bootstrapping significance threshold (adjusted $P$-value $< y$), as well as biased toward the same isoform, denoted as FP $(x, y)$. Then, the FDR in each set of $(x, y)$ was estimated as FP$(x, y)$ divided by the number of real comparisons passing the same criteria.

## 5′UTR sequence feature analysis

To correlate sequence features in the 5′UTR to observed TE difference, we first determined the 5′UTR sequences between the TSSs identified in this study and the start codons annotated in RefSeq. In principle, if there is no splicing between the TSS and start codon, the genomic sequence in between is the 5′UTR sequence; if an intron is constitutively spliced out, the 5′UTR sequence is the concatenation of exonic sequences in between. We reconstructed the splicing patterns in the 5′UTRs, by integrating the RefSeq gene annotation and RNA-seq data derived from the same mouse 3T3 cells, as the splicing site annotation is not complete, in particular for the TSSs outside the gene annotation. For each splicing event (either annotated or detected in RNA-seq), we calculated the percent-spliced-in (PSI) value by counting the number of RNA-seq reads that supported splicing-in or splicing-out. For the RefSeq annotated events, to avoid the uncertainty due to low sequencing coverage, in addition to the real RNA-seq reads, we added 10 pseudo-junction reads for those annotated as constitutive splicing, and 5 splicing-in and 5 splicing-out pseudo-reads for those annotated as alternative splicing. Based on the obtained PSI value, we considered the events with PSI $\leq 0.1$ as constitutively spliced out and PSI $\geq 0.9$ as constitutively spliced in. To avoid uncertainty in determining 5′UTR sequences, isoforms with alternative splicing ($0.1 < PSI < 0.9$) in the 5′UTRs were excluded in sequence feature analysis. Out of 17,033 TSS isoforms, 13,340 were retained. Consequently, 6,536 isoform pairs were retained for comparison, of which 1,025 pairs showed significant TE divergence. With the determined 5′UTR sequences, the upstream ORFs, upstream AUGs (in-frame and out-of-frame), TOP sequences, and hexamers were counted using custom Perl scripts. Local RNA secondary structure minimum free

energy (MFE) and ensemble free energy (EFE) were calculated using RNAfold from the ViennaRNA package version 2.1.9 with option "-p" and otherwise default parameters at a temperature of 37°C (Lorenz *et al*, 2011).

## Independent validation of TSS isoforms and their associated translational efficiency

To validate our findings based on the high-throughput 5′ end sequencing, we used the TeloPrime Full-Length cDNA Amplification kit (Lexogen) to independently determine the 5′ end of capped mRNA. In brief, a gene-specific primer was used to synthesize the complementary DNA (see Table EV6). A double-stranded adapter with a 5′-C overhang that allows for an atypical base pairing with the inverted G of the cap structure was then used for ligation, which can only take place if the RT has reached the 5′ end of the mRNA [Lexogen's unique cap-dependent linker ligation (CDLL)]. After second-strand synthesis, the dsDNA was amplified by a 30-cycled PCR using 5′ Lexogen primer (FP: 5′-TGGATTGATATGTAATAC-GACTCACTATAG) and 3′ gene-specific primers (Table EV6). Amplified products of RNAs from non-ribosomal (pool of free ribosomal, 40S/60S sub-ribosomal fractions) and polysomal fractions (pool of fractions with at least 2 ribosomes) were loaded onto an agarose gel (1%).

## Luciferase reporter assay

To investigate the impact of 5′UTR sequence on translation, longer and shorter versions of 5′UTRs derived from eight genes were PCR-amplified from genomic DNAs, or cDNAs if there is an intron within the 5′UTRs. During PCR, NcoI and BglII restriction sites were introduced to the upstream and downstream of the 5′UTR sequences, respectively. Each 5′UTR fragment was then inserted into the multiple cloning site of the pLightSwitch_5′UTR vector (Active Motif) downstream of an ACTB promoter and upstream of RenSP luciferase reporter ORF. All constructs were validated by Sanger sequencing. Plasmids were transfected into 3T3 cells by using Lipofectamine® 2000 Transfection Reagent (Life Technologies) following the manufacturers' instructions. Luciferase assay was conducted using the LightSwitch Luciferase Assay Reagent™ (Active Motif) and the luciferase activity was measured by Infinite® M200 (Tecan) plate reader and normalized by the absorbance of lysate at 260 nm. Total RNA was extracted from the same lysate using TRIzol® LS Reagent (Life Technologies) and Direct-zol™ RNA Kits (Zymo Research) following the manufacturers' instructions. DNA was removed by in-column DNase I digestion. RT–qPCR was performed to measure the RenSP mRNA level, which was then normalized by the mRNA level of housekeeping gene ActB. Translation efficiency of different constructs was estimated as the normalized luciferase activity divided by normalized RenSP mRNA level.

To validate the effect of putative motifs on translational regulation, ~100-nt sequence stretches containing five copies of specific hexamer motif were synthesized. An AflII site and a BglII site were also included in the 5′ and 3′ ends. As negative control, the sequence stretches containing the reverse complement sequence and the randomly shuffled sequence of hexamer motifs were used, respectively. The test and control sequences were then amplified by PCR. After restriction enzyme digestion, each motif-containing or control sequence stretch was cloned into the multiple cloning site of the pLightSwitch_5′UTR vector. The most upstream motif was 34 nt downstream of 5′ transcript end and the most downstream motif was 45 nt upstream of the start codon. The gap between any two adjacent motif repeats was 4 nt. Translation efficiency of different constructs was measured as described above.

All the primer sequences are listed in Table EV6.

## Initiating ribosome profiling and ORF detection

Mouse NIH3T3 cells were cultured in the same way as for polysome profiling (see above). Harringtonine was added to cell culture at a final concentration of 2 μg/ml. Cells were incubated at 37°C for 120 s. Cycloheximide was then added at cell culture to a final concentration of 100 μg/ml. Cells were immediately lysed in the same way as described for polysome profiling (see above). After lysis, ribosome-protected fragments were collected as described in Ingolia *et al* (2012), with minor modifications. In brief, cell lysate was treated with RNase I at room temperature for 45 min. The nuclease digestion was stopped by adding SUPER-ase·In™ RNase inhibitor (Invitrogen). Monosomes were purified using illustra™ MicroSpin S-400 HR columns (GE Healthcare) following the instruction of ARTseq™ Ribosome Profiling kit (Epicentre). RNA was isolated as described for polysome profiling (see above). rRNA was removed using Ribo-Zero™ Magnetic kit (Human/Mouse/Rat) (Epicentre). The 28- to 32-nt ribosome-protected fragments were purified through 15% (wt/vol) polyacrylamide TBE–urea gel. The size-selected RNA was end-repaired by T4 PNK for 1 h at 37°C followed by heat inactivation at 70°C for 10 min. The dephosphorylated RNA was precipitated by ethanol and then ligated with a preadenylated FTP-3′ adaptor for 2.5 h at room temperature. The ligation product was purified through 15% (wt/vol) polyacrylamide TBE–urea gel and then reverse-transcribed by FTP-RT primer using SuperScript III (Invitrogen) according to the manufacturers' instructions. RT product was ethanol precipitated and further purified through 15% (wt/vol) polyacrylamide TBE–urea gel. Circularization of the RT product was performed in the reaction containing 1× CircLigase buffer, 50 mM ATP, 2.5 mM MnCl$_2$, and 100 U CircLigase (Epicentre) at 60°C for 1 h, and the reaction was heat inactivated at 80°C for 10 min. Circularized cDNA template was amplified by PCR for 12 cycles using the Phusion High-Fidelity DNA Polymerase. The final libraries were sequenced in 1 × 50 nt manner on Illumina HiSeq2000 platform. All the primer and adaptor sequences were listed in Table EV6.

After removing adaptors, sequencing reads that mapped to the reference sequences of rRNA, tRNA, snRNA, snoRNA, and miscRNAs were discarded. The remaining reads were then mapped to the mouse reference genome, allowing up to two mismatches. Reads that were mapped to multiple genomic locations were excluded from further analysis. Then, the data together with published ribosome footprinting data (Shalgi *et al*, 2013) were fed to ORF-RATER (Fields *et al*, 2015) for ORF detection with parameters "--codons NTG" for ORF types, "--minrdlen 28 --maxrdlen 34" for our initiating ribosome profiling data, and "--minrdlen 27 --maxrdlen 34" for published ribosome footprinting data. The detected ORFs were sorted into subtypes, including uORF, annotated ORF, and downstream ORF.

## Gene ontology (GO) enrichment analysis

The gene symbols were mapped to GO terms using R packages GO.db, AnnotationDbi, and org.Mm.e.g.db. GO terms with at least 10 genes from a background set specified in the main text were tested for enrichment in each of studied gene sets using the GOseq method provided in the R package "goseq" (Young *et al*, 2010). The raw *P*-values were then adjusted by using the Benjamini–Hochberg (BH) procedure.

## Construction of quantitative models explaining the TE differences between alternative TSS isoforms

To understand the individual and combinatory contribution of different sequence features to the TE difference observed between alternative TSS isoforms, we built nonlinear multivariable regression models using the multivariate adaptive regression splines (MARS) approach (Friedman, 1991), which can automatically select independent variables and model the nonlinearities between the selected independent variables and the responding variable. The modeling analysis was performed in R (version 3.2.2) with the R package "earth" (version 4.4.3). The function earth with parameters "degree = 1, penalty = 2, thresh = 0.001, fast.k = 0, fast.beta = 0" was used to build models. The parameter "degree" defined the maximum degree of interaction between variables, and the value 1 meant to build additive models with no interaction terms allowed. The parameter "penalty" was the penalty in generalized cross-validation, and the value 2 was the default setting for degree = 1. The setting "thresh = 0.001" was one of the computation termination criteria, tuning between computing time and model performance. Setting "fast.k = 0, fast.beta = 0" disabled fast calculation. The function predict was used to predict TE divergence between TSS isoforms on test data.

To assess the individual contribution of sequence features, we built quantitative models for each feature separately and took the variance of observed TE divergence explained by each model as their individual contribution. In the analysis of combinatory contribution of sequence features, we sequentially added sequence features to models in the descending order of their individual contribution and measured their cumulative contribution as the variance explained by the model combining these sequence features. Delta cumulative contribution was calculated as the additional variance explained by adding the specific sequencing feature to the combinatory models. Delta cumulative contribution was used to estimate the extent of additional information gained by considering one more feature given the interdependence between different features.

## Data availability

The raw sequencing data have been submitted to NCBI GEO database under accession number GSE78241. Analysis scripts are available at https://github.com/sunlightwang/CAPTRE and as Computer Code EV1.

**Expanded View** for this article is available online.

## Acknowledgements

We thank Madlen Sohn, Mirjam Feldkamp, and Claudia Langnick for their excellent technical assistance. As part of the Berlin Institute for Medical Systems Biology at the MDC, the research group of Wei Chen is funded by the Federal Ministry for Education and Research (BMBF) and the Senate of Berlin, Berlin, Germany (BIMSB 0315362A, 0315362C).

## Author contributions

XW, JH, and WC conceived and designed the project. JH did the experiments with help from CQ. XW analyzed the data. XW, JH, and WC wrote the manuscript. All authors read and approved the final manuscript.

## Conflict of interest

The authors declare that they have no conflict of interest.

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
