## [Review Process File · Molecular Systems Biology]

Pervasive isoform-specific translational regulation via alternative transcription start sites in mammals

Xi Wang, Jingyi Hou, Claudia Quedenau and Wei Chen

Corresponding author: Wei Chen, Max-Delbrück-Centrum für Molekulare Medizin and South University of Science and Technology of China

Review timeline:

Submission date:	10 March 2016
Editorial Decision:	05 April 2016
Revision received:	17 June 2016
Accepted:	21 June 2016

Editor: Maria Polychronidou

Transaction Report:

1st Editorial Decision

05 April 2016

Thank you again for submitting your work to Molecular Systems Biology. We have now heard back from the three referees who agreed to evaluate your study. As you will see below, overall the reviewers think that the study represents a valuable contribution to the field. However, they raise a number of concerns, which should be carefully addressed in a revision of the manuscript. The reviewers' recommendations are rather clear so there is no need to repeat the points listed below.

REFeree COMMENTS

Reviewer #1:

Using polysome profiling in combination with 5' cap-dependent sequencing, Wang, Hou, and colleagues determined the translational differences conferred by alternative 5' UTRs. By pairing isoforms that are presumably otherwise identical, they were able to directly test the roles of 5' UTR features in over 4,000 cases in NIH 3T3 cells. Consistent with previous studies comparing 5' UTRs between transcripts that are not otherwise identical, longer 5' UTRs tend to reduce translational capacity. Specific features that contribute to this reduction are upstream ORFs, stable RNA structures, and pyrimidine tracts, as well as certain sequence motifs.

The experimental procedures and statistical methods presented are sound and clearly presented. The pairwise comparison of translational status between isoforms is a well-controlled system for determining effects of 5' UTR features, and although it has previously been done in yeast (Arriberre & Gilbert, 2013), this study presents a far more thorough analysis. Results here confirm previous knowledge about the effect of 5' UTR features on translation. Unfortunately only minor insight is

gained beyond the previous knowledge: namely, the contribution of each feature to translational regulation in this cell line, and the identification of new sequence motifs that decrease translation.

Major concern:

The scatter plot analysis comparing the authors' TE data and ribosome profiling TE data is highly non-linear (EV2, panel C). This suggests that the 5' sequencing analysis is not a quantitative approach, making the isoform TE differences challenging to interpret. To address this the authors could perform isoform abundance quantification by RNA-seq analysis of the fractions and analyze the abundance of different 5' UTRs as defined by the 5' end sequencing TSS mapping. However, as the authors indicate, this is a challenging analysis to perform using short-read RNA-seq. Comparing RNA-seq data with the 5' end sequencing data (for one or all samples) would allow a more precise measure of how quantitative the approach is. So at a minimum, it is important to determine how accurate the measures of isoform TE differences are and how the non-linear nature of the data may impact their conclusions.

Minor points:

The authors state that recent studies (2011, 2012) have demonstrated comparable contribution of transcription and translation to cellular protein abundance. However, even more recent studies have shown a dominant effect of transcription on protein abundance (Jovanovic et al., 2015; Li et al., 2014).

In Figure 1D there are gray peaks that are very hard to see and not described in the figure legend. I assume these are reads that were not counted as part of TSSs. An analysis of the fraction of mapped reads that are considered within TSSs would be informative for judging the quality of the data.

Typically TE is calculated from ribosome footprint and RNA-seq reads, giving a unitless value. Here TE is calculated from different types of data and expressed as ribosomes/kb. Another term, such as "ribosome density" may help avoid confusion.

In the leave-one-fraction-out analysis, leaving the free fraction appears to significantly reduce the FC range (Figure EV2E). This should be addressed.

The hexamers identified that significantly correlate with TE divergence should be listed, as this is one of the main novel findings of the study.

It is not clear why the individual contribution of hexamers shown in Figure 6A is "n.a." Based on the cumulative change when accounting for hexamers they have a significant contribution to translation regulation.

References

Jovanovic, M. et al. Immunogenetics. Dynamic profiling of the protein life cycle in response to pathogens. *Science* 347, 1259038, doi:10.1126/science.1259038 (2015).

Li, J.J. et al. System wide analyses have underestimated protein abundances and the importance of transcription in mammals. *PeerJ* 2, e270. (2014)

Reviewer #2:

Summary:

In this study authors investigated genome-wide mRNA isoform-specific translational control in mammalian cells. They used polysome profiling with high-throughput mRNA 5' end sequencing and measured translation efficiency of mRNA isoforms with different transcription start sites (TSS) in mouse NIH 3T3 cells. Around 20% of genes with multi-TSS showed divergent translation efficiency between alternative TSS isoforms. Authors developed a quantitative model and could explain over half of this variance between TSS isoforms, describing translation regulatory roles of existing and novel sequence elements in 5' UTR. They claim it to be the first study to report global impact of 5' UTR isoforms on translational control and first to provide genome-wide in vivo evidence that stable mRNA structures in 5' UTR reduces translation efficiency in mammals.

General Remarks:

This study elevates the general understanding of alternative TSS-based translational control in murine fibroblast. Some previous studies had noted the effect of regulatory sequences in 5' UTR on translation of particular genes, but here it is studied at genome-wide scale in mammalian cells. As stated by authors, it needs further studies to confirm that different cells and tissues share the translation regulation rules reported here.

However, current study utilizes polysome profiling to measure the translation efficiency, which is widely used for this purpose but sometimes may produce overestimated result. It is known that presence of some high order structures like lipid rafts, pseudo-polysomes can be erroneously regarded as polysomes [Thermann R et al. Nature, 2007] and some detergents (e.g.: Triton X-100) absorbing at 254 nm can mask 40/60 S peaks. In addition, number of ribosomes on an mRNA does not always point to active translation [Graber TE, et al. Proc Natl Acad Sci USA. 2013]. Here, authors could only explain 50% of translational variation between TSS isoforms, which may be due to consideration of only 5' UTR features and presence of some false positive readings.

Nonetheless, a high number of TSSs are newly identified in this study expanding the current TSS repertoire and translation regulatory sequence features of 5' UTR found here can be helpful for the researchers working in the field of transcriptomics/proteomics. Some major changes are needed to improve the content.

Major Points:

1. It is not clear whether PCR duplicates were taken into account during filtering of mRNA 5' end sequencing reads.

2. In Figure 1D and 2B, the range of Y-axis (read coverage) is different for each fraction, for better comparison of reads across fractions, it's recommended to fix the range for all the fractions and mention the unit of read coverage. In Figure 1D, the gene structure of Cnot1 seems incomplete and also gene structure for alternative TSS isoform for this gene is missing. For the sake of readers' interest, it will be good to briefly state about the function of these genes.

3. Authors randomly chose four genes for the confirmation of the observed isoform-specific TE (shown in Figure 2B), it will be nice to mention their biological function or if they are reported in the literature. In the same figure, in case of Ndufb11 agarose gel image, there is a faint band between distal and proximal TSS isoforms, indicating presence of intermediate TSS isoform but there are no reads representing this in either of free and Poly9+ fractions, why?

In case of Ube4b, there are many small reads in gray flanking from distal TSS to downstream of proximal TSS isoform, no explanation for these are given. Here, alternative TSS isoforms cumulative reads and their band intensity in agarose gel is not concordant (Proximal TSS in blue have around 30 times more reads in poly9+ compared to free fraction, but this difference is not visible in the gel image. Moreover, if you compare reads and band intensities of proximal and distal TSS in poly9+, these don't show correlation). The same goes for Nedd8 and Ssu72. The Ssu72 distal TSS (in red) have more reads in poly9+, but the gel image is suggesting just the opposite.

So, I don't agree with the following conclusion: Page 9, third line from bottom

"More importantly, the relative abundance of TSS isoforms in both non-ribosomal and polysomal fractions agreed to that determined by our global approach." Please correct the statements related to this experiment.

To confirm the TE variation between TSS isoforms, it would be better to transiently express some of the isoforms with high TE divergence in a cell line (lacking these isoforms) and checking mRNA and protein level by qRT-PCR and western blot respectively.

4. It is mentioned in the abstract and on Page 8 that 745 alternative TSS showed significant differential TE, but in the last line on Page 10 (supported by Figure 3), it says: "Among the 1,025 isoform pairs with significant TE difference....."

Please comment on these two different numbers of TSS isoforms with differential TE.

5. Recently a novel class of antisense lncRNAs (SINEUP) has discovered which binds to its sense counterpart mRNA in 5' head-to-head manner and increases translation of mRNA [Carrieri et al. Nature 2012]. Have you found such antisense lncRNAs associated with translating mRNA or expected binding sites for these in your data? Notably, as shown in Figure 2C, the translation difference between short and long 5'UTR of Ndufb11 is very high, have you checked if this is due to presence of IRES or any SINEUP binding site in the short form or the longer form have some miRNA binding sites? It will be interesting and informative to include such analysis and discuss possibility of IRES, miRNA and SINEUP binding sites.

Are the eight genes shown in this figure belong to a particular pathway, in other words, is there any interconnection between them? Please comment.

6. It is demonstrated here that 5' cap-adjacent stable RNA secondary structures inhibit translation; did you find any consensus secondary structure or any part of structure common in multiple transcripts? Please provide information of such secondary structures with their minimum free energy.

7. In the "novel sequence motifs associated with isoform-specific translation" analysis (Page 14), the reason behind giving preference to hexamer motifs and choosing AAUCCC and CAAGAU motifs for validation is not given, further explanation is required. Do these two motifs have similarity to any known motifs? Five copies of these motifs were used, explain on what basis this number of repeats decided, also relative distance of hexamers from the start codon and 5' cap and gap between two hexamer repeats is not mentioned. Please provide this information.

8. As it's well known that Kozak sequence plays an important role in efficient translation, I suggest to analyze the sequence data for the presence/absence of Kozak sequence. It would be interesting to know whether TSS isoform pairs varying in TE also differ in terms of Kozak sequence.

9. How many isoforms have multiple translation regulatory sequence features in 5' UTR mentioned in Figure 6A? It is not explained what delta cumulative is representing here and how it is calculated, please describe it.

10. In figure EV2 C and D the unit of Y-axis is not given. In fact, I doubt that translation efficiency calculated here can be compared with the ribosome footprinting data (Eichhorn et al, 2014) and proteomics data (Schwanhäusser et al, 2011). As these studies have different measures of TE and different methods were employed to calculate it. Moreover, in my understanding, for the ribosome footprinting Eichhorn et al used miRNA-inducible NIH3T3 cells, and to collect proteomics data, Schwanhäusser et al cultured NIH3T3 cells in SILAC medium which is different from current study. Different culture conditions and passage number may affect the result. So, in my opinion, statements related to this analysis should be removed from the manuscript.

Minor Points:

1. Correct the grammatical errors, such as :

Please change single quote (') to prime sign (') throughout the manuscript. For instance, change 5' UTR to 5' UTR.

In Figure 2C Y-axis, correct spelling of Luciferase.

2. In Figure EV1 B, please describe what Y-axis is and what different colors are representing.

3. "Takahashi et al, 2012" reference is missing from the list.

Reviewer #3:

Wang & Hou et al. show that variation in transcription start sites (TSSes) of mammalian transcripts (in cell culture) can affect their translational efficiency, as measured by their position on polysome profiles. By using CAGE-based techniques to define 5' ends of transcripts, the authors build on previous work to show that sequence elements and features in 5' leaders* can impact downstream translation (*we will use 5' leader instead of 5' UTRs, since many 5' leaders are translated over their uORFs).

As noted by the authors, this work is similar to recently-published work by Floor and Doudna (10.7554/eLife.10921), as well as Dieudonne et al. (10.1186/s12864-015-2179-8; not cited). However, a key distinction is that Wang & Hou et al. use experimental techniques to identify 5' ends de novo and quantify their use, rather than relying on the quantification of 5' ends of existing annotations from RNA-Seq data. The authors do show that this de novo annotation and quantification is important, as many of these identified 5' ends do not coincide with annotated 5' ends.

All things considered, we recommend this manuscript for publication. Overall, the work is well-documented, and high throughput experiments are independently validated. While we would like to see some analyses more explicitly explained, we do not see the need for further experiments. The findings of this paper are no surprise to anyone in the field, but have not been rigorously demonstrated prior to this work, particularly in mammalian cells. The work would be a meaningful addition to the existing literature.

Major points:

1. Please define exactly what is meant by the "gross 5'-UTR region." This is an important concept and does not appear to be defined in either the results or the methods sections.
2. In-frame upstream AUGs should be subdivided into AUGs that are merely N-terminal extensions of the CDS, and bona fide uORFs that may be in-frame with, but terminate before the CDS (Figs 4B, C). This would help determine why in-frame uAUGs differ from out-of-frame uAUGs in their effects on downstream translation.
3. It is also not clear what "individual", "cumulative" and "delta cumulative" mean in Fig 6A: please interpret this for the reader (who should not be expected to fire-up the R 'earth' package to do so).
4. Please justify the use of the MARS approach instead of simpler linear models (with appropriately transformed parameters). Similarly, the specific parameters used should be enumerated more clearly.
5. Cycloheximide pre-treatment prior to cell lysis (as was done in the experiments in this work) causes a known experimental artifact: because it stalls elongating ribosomes, it tends to cause a 5' pile-up of ribosomes, which can be observed in ribosome profiling data. This artifact can potentially confound polysome profiling data, as shorter ORFs may be predicted to have disproportionately more ribosomes (per unit length), as would polycistronic transcripts (i.e. transcripts with multiple uORFs). Unfortunately, the established protocols for polysome profiling do not account for this. We recommend checking if this is an issue, by plotting calculated associated ribosomes against the length of the CDS, perhaps for the transcripts without uORFs / uAUGs in the 5' leader. If 5' pile-up were an issue, one might expect a significant residual number of associated ribosomes as the length of the CDS goes to 0. Should this y-intercept be substantial, it would need to be accounted for in the calculation of translational efficiency.
6. Gene Ontology enrichment analyses based on sequencing data are necessarily biased by gene expression levels, as highly expressed genes have more reads and so pass statistical thresholds more readily. This problem can be ameliorated by using a method such as Goseq (PMID 20132535). The authors should repeat their enrichment analysis using an appropriate bias correction.
7. The sequencing data generated in this study must be deposited in a public repository (e.g. SRA or ENA).

Minor points:

1. Spearman's rank correlation coefficient is denoted by the Greek letter " ρ ," not "R."
2. It would be useful to show the distribution and dynamic range of translational efficiency observed across various genes, particularly as a comparison to similar measures of translational efficiency typically calculated from ribosome profiling data.
3. While they are likely to be well-correlated with each other in most cases, we suggest the use of ensemble free energy (EFE) over minimum free energy (MFE) as the most relevant value for assessing secondary structure stability. This can be evaluated using ViennaRNA's `pf_fold` function, or RNAfold -p (Figs 5A, B).
4. Technically, ORFs are defined purely by sequence: the transcript segment defined by a start codon on the 5' end, and an in-frame stop codon on the 3' end. Thereafter, uORFs can be classified (by some threshold) as translated or untranslated. This distinction should be made for Figs 4D-G.
5. Given that arbitrary thresholds and significance levels are used to classify genes as multiply-initiated (4153/9951) or translationally-divergent (745/4153), we would qualify such statements appropriately, e.g. 4,153 of 9,951 expressed genes showed significant initiation at multiple TSSs.
6. Scatter plots are over-plotted; use some transparency in the points so that the density at different regions is better appreciated.
7. The genes that were individually validated in Fig 2C should be highlighted accordingly on Fig 6B. They could be depicted as horizontal line segments, where both observed values (from individual experiments and high-throughput data) are joined.
8. Fig 3C is redundant (all the information is already captured in Fig 3A).
9. While this may not be necessary for publication, we highly encourage the publication and/or public deposition of all code and data tables necessary for the analyses presented in this work.

Citation issues:

1. More recent work quantifying the relative contribution of transcription and translation should be cited as well: 10.1126/science.1259038 and 10.7717/peerj.270, particularly because they suggest that transcriptional control plays a larger contribution (see 10.1126/science.aaa8332)

2. Another key prior work that has investigated the contribution of isoforms to translation (albeit in yeast) is 10.1038/nature12121. It should be referenced also.
3. 10.1186/s12859-014-0380-4 could be a useful citation for explaining the lack of effect in non-AUG codons initiation.

1st Revision - authors' response

17 June 2016

Reviewer #1:

Using polysome profiling in combination with 5' cap-dependent sequencing, Wang, Hou, and colleagues determined the translational differences conferred by alternative 5' UTRs. By pairing isoforms that are presumably otherwise identical, they were able to directly test the roles of 5' UTR features in over 4,000 cases in NIH 3T3 cells. Consistent with previous studies comparing 5' UTRs between transcripts that are not otherwise identical, longer 5' UTRs tend to reduce translational capacity. Specific features that contribute to this reduction are upstream ORFs, stable RNA structures, and pyrimidine tracts, as well as certain sequence motifs.

The experimental procedures and statistical methods presented are sound and clearly presented. The pairwise comparison of translational status between isoforms is a well-controlled system for determining effects of 5' UTR features, and although it has previously been done in yeast (Arribere & Gilbert, 2013), this study presents a far more thorough analysis. Results here confirm previous knowledge about the effect of 5' UTR features on translation. Unfortunately only minor insight is gained beyond the previous knowledge: namely, the contribution of each feature to translational regulation in this cell line, and the identification of new sequence motifs that decrease translation.

R: We thank the reviewer for the positive comments on our experiment design and data analyses.

Major concern:

The scatter plot analysis comparing the authors' TE data and ribosome profiling TE data is highly non-linear (EV2, panel C). This suggests that the 5' sequencing analysis is not a quantitative approach, making the isoform TE differences challenging to interpret. To address this the authors could perform isoform abundance quantification by RNA-seq analysis of the fractions and analyze the abundance of different 5' UTRs as defined by the 5' end sequencing TSS mapping. However, as the authors indicate, this is a challenging analysis to perform using short-read RNA-seq. Comparing RNA-seq data with the 5' end sequencing data (for one or all samples) would allow a more precise measure of how quantitative the approach is. So at a minimum, it is important to determine how accurate the measures of isoform TE differences are and how the non-linear nature of the data may impact their conclusions.

R: In the previous Figures EV2C and D, we plotted TE values measured by polysome profiling versus log₂-transformed TE values based on ribosome footprinting and proteomics, respectively. In the revised manuscript, we have redrawn both scatterplots in log scale for both axes. Now no non-linearity is observed (see new Fig. EV2C,D).

To assess the accuracy of the 5'-end sequencing on RNA abundance quantification, we compared the gene expression estimated based on 5'-end sequencing to that on regular RNA-seq. The values from the two technologies correlated well (Figure R1). Importantly, the correlation is even better for genes with single TSS isoform, suggesting that the divergence between the two approaches was at least partially due to the well-known challenge of regular RNA-seq in quantifying the expression of multi-isoform genes.

Figure R1. Scatterplots comparing RNA abundance quantified using 5'-end sequencing (Y-axis) to that using regular RNA-seq (X-axis) for all genes (left) or for only the genes with single TSS isoform (right).

Minor points:

The authors state that recent studies (2011, 2012) have demonstrated comparable contribution of transcription and translation to cellular protein abundance. However, even more recent studies have shown a dominant effect of transcription on protein abundance (Jovanovic et al., 2015; Li et al., 2014).

R: We appreciate that the reviewer raised this question. In our opinion, it is still debatable which step is predominant in determining cellular protein abundance. Likely, the answer to this question is also dependent on cell types and conditions. In the revised manuscript we have added all relevant references and adjusted the sentence in the Introduction section.

In Figure 1D there are gray peaks that are very hard to see and not described in the figure legend. I assume these are reads that were not counted as part of TSSs. An analysis of the fraction of mapped reads that are considered within TSSs would be informative for judging the quality of the data.

R: We would like to thank the reviewer for pointing out this issue. Indeed, the gray bars are sequencing reads mapped outside of the TSS peak regions that we identified. In the 5' end sequencing data across the seven fractions, on average 85% of reads were located within TSS regions, indicating the high quality of our data. In the revised manuscript, we have clarified this in the figure legends and the Materials and Methods section.

Typically TE is calculated from ribosome footprint and RNA-seq reads, giving a unitless value. Here TE is calculated from different types of data and expressed as ribosomes/kb. Another term, such as "ribosome density" may help avoid confusion.

R: We thank the reviewer for this suggestion and have adjusted the figure labels accordingly.

In the leave-one-fraction-out analysis, leaving the free fraction appears to significantly reduce the FC range (Figure EV2E). This should be addressed.

R: To address this issue, we estimated the relative abundance of different TSS isoforms across the seven fractions (i.e., isoform abundance in one fraction divided by the overall abundance summing up from all fractions), and calculated the difference in the relative abundance between any pairs of alternative isoforms from the same gene for the seven fractions separately. As shown in Figure R2, the pair-wise difference spans the widest range in the free fraction, indicating the largest contribution of the free fraction in calculating TE divergence. Therefore, leaving the free fraction out reduced the divergence range.

Figure R2. Boxplots showing the distribution of the difference in the relative abundance between alternative isoform pairs in the seven fractions.

The hexamers identified that significantly correlate with TE divergence should be listed, as this is one of the main novel findings of the study.

R: In the revised manuscript, we have added Table EV4 to list all the hexamer motifs that were significantly associated with isoform-specific translation.

It is not clear why the individual contribution of hexamers shown in Figure 6A is "n.a." Based on the cumulative change when accounting for hexamers they have a significant contribution to translation regulation.

R: In the previous manuscript, we felt that it was impossible to list all the individual contribution of each hexamer in the figure. Now in the revised manuscript, we have added the contribution of all hexamers to Figure 6A.

References Jovanovic, M. et al. Immunogenetics. Dynamic profiling of the protein life cycle in response to pathogens. *Science* 347, 1259038, doi:10.1126/science.1259038 (2015).

Li, J.J. et al. System wide analyses have underestimated protein abundances and the importance of transcription in mammals. *PeerJ* 2, e270. (2014)

Reviewer #2:

Summary:

In this study authors investigated genome-wide mRNA isoform-specific translational control in mammalian cells. They used polysome profiling with high-throughput mRNA 5' end sequencing and measured translation efficiency of mRNA isoforms with different transcription start sites (TSS) in mouse NIH 3T3 cells. Around 20% of genes with multi-TSS showed divergent translation efficiency between alternative TSS isoforms. Authors developed a quantitative model and could explain over half of this variance between TSS isoforms, describing translation regulatory roles of existing and novel sequence elements in 5' UTR. They claim it to be the first study to report global impact of 5' UTR isoforms on translational control and first to provide genome-wide in vivo evidence that stable mRNA structures in 5' UTR reduces translation efficiency in mammals.

General Remarks:

This study elevates the general understanding of alternative TSS-based translational control in murine fibroblast. Some previous studies had noted the effect of regulatory sequences in 5' UTR on

translation of particular genes, but here it is studied at genome-wide scale in mammalian cells. As stated by authors, it needs further studies to confirm that different cells and tissues share the translation regulation rules reported here.

However, current study utilizes polysome profiling to measure the translation efficiency, which is widely used for this purpose but sometimes may produce overestimated result. It is known that presence of some high order structures like lipid rafts, pseudo-polysomes can be erroneously regarded as polysomes [Thermann R et al. Nature, 2007] and some detergents (e.g.: Triton X-100) absorbing at 254 nm can mask 40/60 S peaks. In addition, number of ribosomes on an mRNA does not always point to active translation [Graber TE, et al. Proc Natl Acad Sci USA. 2013]. Here, authors could only explain 50% of translational variation between TSS isoforms, which may be due to consideration of only 5' UTR features and presence of some false positive readings. Nonetheless, a high number of TSSs are newly identified in this study expanding the current TSS repertoire and translation regulatory sequence features of 5' UTR found here can be helpful for the researchers working in the field of transcriptomics/proteomics. Some major changes are needed to improve the content.

R: We thank the reviewer for the positive remarks.

Major Points:

1. It is not clear whether PCR duplicates were taken into account during filtering of mRNA 5' end sequencing reads.

R: We did not collapse reads with the same nucleotide sequence. Since mRNA transcripts derived from a gene locus are often initiated from a precise and narrow genomic window, they can generate 5'-end sequencing reads with the same sequences. It is in particular the case for highly expressed transcripts. Thus, we do not believe the reads of identical sequences represent PCR duplicates.

2. In Figure 1D and 2B, the range of Y-axis (read coverage) is different for each fraction, for better comparison of reads across fractions, it's recommended to fix the range for all the fractions and mention the unit of read coverage. In Figure 1D, the gene structure of *Cnot1* seems incomplete and also gene structure for alternative TSS isoform for this gene is missing. For the sake of readers' interest, it will be good to briefly state about the function of these genes.

R: In Figure 1D and 2B, we illustrated the relative abundance of different TSS isoforms across fractions. Due to the large abundance difference across functions, to fix the range would make the figure unreadable. To avoid confusion, in the revised manuscript, we have clarified in the figure legend about the variable range of Y-axis. For *Cnot1*, we have revised Figure 1D by adding more exons including the start of its ORF, and added the gene structure for alternative isoform. As suggested by the reviewer, we have listed the function of these genes in Table EV3.

3. Authors randomly chose four genes for the confirmation of the observed isoform-specific TE (shown in Figure 2B), it will be nice to mention their biological function or if they are reported in the literature. In the same figure, in case of *Ndufb11* agarose gel image, there is a faint band between distal and proximal TSS isoforms, indicating presence of intermediate TSS isoform but there are no reads representing this in either of free and Poly9+ fractions, why? In case of *Ube4b*, there are many small reads in gray flanking from distal TSS to downstream of proximal TSS isoform, no explanation for these are given. Here, alternative TSS isoforms cumulative reads and their band intensity in agarose gel is not concordant (Proximal TSS in blue have around 30 times more reads in poly9+ compared to free fraction, but this difference is not visible in the gel image. Moreover, if you compare reads and band intensities of proximal and distal TSS in poly9+, these don't show correlation). The same goes for *Nedd8* and *Ssu72*. The *Ssu72* distal TSS (in red) have more reads in poly9+, but the gel image is suggesting just the opposite. So, I don't agree with the following conclusion: Page 9, third line from bottom "More importantly, the relative abundance of TSS isoforms in both non-ribosomal and polysomal fractions agreed to that determined by our global approach." Please correct the statements related to this experiment.

R: As stated above, we have added the description of gene functions in Table EV3.

In the case of *Ndufb11*, the band below the distal TSS in the Agarose gel image represented an alternative splicing event within the region between the proximal and distal TSSs. This splicing

would remove an 88-nt region for a minor fraction of transcripts starting at the distal TSS. However, the alternative splicing cannot be distinguished based only on the 5'-end sequencing reads, therefore only two types of TSSs (proximal and distal) were observed in the genome browser plot. In the revised manuscript, we have clarified this in the figure legend.

In genome browser plots, the gray bars are sequencing reads mapped outside of the TSS peak regions, which were identified based on the 5'-end sequencing data pooled from all seven fractions. In the free fraction of *Ube4b*, the read coverage was relatively low, which made the background apparent.

In Figure 2B, instead of absolute RNA isoform abundance in different fractions, we aimed to compare, between the two approaches, the relative abundance of these isoforms across fractions, i.e.

$$\frac{T_{\text{dist,poly}}/T_{\text{prox,poly}}}{T_{\text{dist,nonribo}}/T_{\text{prox,nonribo}}}, \text{ where } T_{\text{dist,poly}} \text{ and } T_{\text{prox,poly}}$$

$T_{\text{dist,poly}}/T_{\text{prox,poly}}T_{\text{dist,nonribo}}/T_{\text{prox,nonribo}}$, where $T_{\text{dist,poly}}$ and $T_{\text{prox,poly}}$ represent the isoform abundance of distal and proximal TSSs in the polysomal fraction, respectively; $T_{\text{dist,nonribo}}$ and $T_{\text{prox,nonribo}}$ represent the isoform abundance in the non-ribosomal fraction. We quantified the gel intensity of bands derived from distinct TSS isoforms in Figure 2B, and calculated the ratio according to the above formula. The values correlated very well to those calculated based on 5'-end sequencing data (Figure EV4). In the revised manuscript, we have made it clear and added the figure as Figure EV4. In our opinion, it is difficult to directly compare the absolute gel intensity and read coverage in Figure 2B, because (i) for inter-fraction comparisons, we did not start with the same amount of RNA from different fractions in the 5'-end-sequencing and the validation experiments; and (ii) for intra-fraction comparisons, DNA fragments with different length show different band intensity even when present in equimolar amounts, because shorter DNA fragments bind less ethidium bromide.

To confirm the TE variation between TSS isoforms, it would be better to transiently express some of the isoforms with high TE divergence in a cell line (lacking these isoforms) and checking mRNA and protein level by qRT-PCR and western blot respectively.

R: We agree with the reviewer that such validation is important. Indeed, to more directly demonstrate that alternative 5'UTR sequences are able to drive the observe TE divergence, we have performed luciferase reporter assay to validate the impact of 5'UTRs in translational regulation for individual genes. As shown in Figure 2C, seven out of eight isoform pairs showed significant differential TE biased towards the same isoforms as observed in the global analysis.

4. It is mentioned in the abstract and on Page 8 that 745 alternative TSS showed significant differential TE, but in the last line on Page 10 (supported by Figure 3), it says: "Among the 1,025 isoform pairs with significant TE difference....." Please comment on these two different numbers of TSS isoforms with differential TE.

R: We apologize for this confusion. In total, we detected 1,618 TSS-isoform pairs in 745 genes with significant TE difference. Please note that one gene with >2 TSSs can have more than one TSS-isoform pair. In the sequence feature analysis, to avoid unambiguously determined 5'UTRs, we only focused on genes without alternative splicing within 5'UTRs (see Materials and Methods). Therefore, among the 1,618 TSS-isoform pairs with significant TE divergence, we only focused on 1,025 pairs where each 5'UTR isoform was unambiguously determined. To avoid such confusion, we have adjusted the relevant sentences in the revised manuscript accordingly.

*5. Recently a novel class of antisense lncRNAs (SINEUP) has discovered which binds to its sense counterpart mRNA in 5' head-to-head manner and increases translation of mRNA [Carrieri et al. Nature 2012]. Have you found such antisense lncRNAs associated with translating mRNA or expected binding sites for these in your data? Notably, as shown in Figure 2C, the translation difference between short and long 5'UTR of *Ndufb11* is very high, have you checked if this is due to presence of IRES or any SINEUP binding site in the short form or the longer form have some miRNA binding sites? It will be interesting and informative to include such analysis and discuss possibility of IRES, miRNA and SINEUP binding sites.*

R: *Ndufb11* showed the largest TE difference between 5'UTR isoforms among the eight genes, likely due to the largest length difference between the long and short 5'UTRs. As suggested by the reviewer, we checked IRES, miRNA target sites, and SINEUP binding sites in the alternative 5'UTR isoforms of *Ndufb11*, but did not find any occurrence of IRES nor target sites for the 100 most highly expressed miRNAs, and none were overlapping with any 'SINEUP' antisense lncRNAs. When extending the investigation to the global level, no association was observed between the appearance of any abovementioned sequence features in the divergent 5'UTR regions of isoform pairs and their TE differences, as explained below in more detail.

1) IRES.

Since there are no common sequences or structural motifs shared among the currently identified cellular IRES elements, we cannot directly use bioinformatics tools to predict IRES in 5'UTRs. Therefore we collected 85 IRES sequences with length less than 1kb from the IRES database (<http://iresite.org/>) (Mokrejs *et al*, 2010), and aligned these sequences to each gene's 5'UTR sequences. We only kept alignments matching more than 90% of the IRES sequences, and this resulted in only a few genes whose 5'UTR sequences containing potential IRESs (*Cttna3*, *Fmr1*, *Hif1a*, *Mnt*, *Myc*, *Runx1t1*). We then checked for TE divergence between isoform pairs in these genes, but did not observe any significant association between such IRES sequence presence and TE divergence (P=0.97; Mann–Whitney U test).

2) miRNA target sites.

Although miRNA target sites can be predicted within 5'UTRs of endogenous mRNAs and it has been reported that miRNA binding sites in 5'UTRs may activate translation in individual cases (Ørom *et al*, 2008), they are less frequent and do not have extensive functional effects as observed for those located in 3'UTRs (Farh *et al*, 2005; Lewis *et al*, 2003; Lim *et al*, 2005). Nevertheless, following the reviewer's suggestion, we examined the correlation between miRNA target site appearance in 5'UTR and TE divergence. We estimated each miRNA's expression level in 3T3 cells based on the small RNA-seq data (GSM774079). For the 100 most abundant miRNAs, we scanned each 5'UTR for their target sites, and correlated their appearance in divergent 5'UTR regions with TE difference. We found no evidence to show that the appearance of miRNA target sites in 5'UTRs had impact on TE regulation (P=0.17; Mann–Whitney U test).

3) SINEUP binding sites.

SINEUP is a very interesting new class of antisense long non-coding RNAs that stimulate translation of their sense mRNAs. However, to the best of our knowledge, currently there are no databases for SINEUP binding sites available. According to previous publications (Carrieri *et al*, 2012; Zucchelli *et al*, 2015), we searched for expressed, anti-sense, non-coding, SINE/B2 containing transcripts (SINEUP RNA) overlapping with 5'UTR isoforms in 3T3 cells. The non-coding RNA annotation (NONCODE 2016) was downloaded from <http://www.noncode.org/> (Zhao *et al*, 2016) and SINE/B2 annotation was retrieved from RepeatMasker outputs downloaded from UCSC genome browser (<http://genome.ucsc.edu/>). After filtering out non-expressed ncRNA based on our 5'end-sequencing data, we correlated the differential association (i.e. genomic position overlapping) of SINEUP RNA in isoform pairs to the TE divergence, but no significant correlation (P=0.22; Mann–Whitney U test) was observed.

Are the eight genes shown in this figure belong to a particular pathway, in other words, is there any interconnection between them? Please comment.

R: We randomly picked up the eight genes for validation, and there is no interconnection between them. In the revised manuscript, we have added the description of gene functions in Table EV3.

6. It is demonstrated here that 5' cap-adjacent stable RNA secondary structures inhibit translation; did you find any consensus secondary structure or any part of structure common in multiple transcripts? Please provide information of such secondary structures with their minimum free energy.

R: In 5'-cap stable RNA structure analysis, we set the MFE threshold to be -30 kcal/mol. Examining these 50-nt RNA fragments with MFE less than -30 kcal/mol, as expected, we found that they in general formed hair-pin structures. In the revised manuscript, we have added two examples in Figure EV6A,B with their secondary structures.

7. In the "novel sequence motifs associated with isoform-specific translation" analysis (Page 14), the reason behind giving preference to hexamer motifs and choosing AAUCCC and CAAGAU motifs for validation is not given, further explanation is required. Do these two motifs have similarity to any known motifs? Five copies of these motifs were used, explain on what basis this number of repeats decided, also relative distance of hexamers from the start codon and 5' cap and gap between two hexamer repeats is not mentioned. Please provide this information.

R: As often used in other studies, we chose hexamer for motif analysis due to their optimal occurrence frequency to achieve high sensitivity and specificity. We chose the two candidates for validation with the consideration of their adjusted p-values, effect size, occurrence frequency in 5'UTRs (see Table EV4). The motif CAAGAU was the one with relatively large effect size and high occurrence frequency, and in contrast, the motif AAUCCC was chosen to represent motifs with weaker effect. Indeed, the reporter assay demonstrated that CAAGAU held stronger repressive effect than AAUCCC. Neither of the two motifs matched to known RNA-binding-protein binding motifs annotated in RBPDB (<http://rbpdb.cbr.utoronto.ca>) or RBPmap (<http://rbpmap.technion.ac.il>).

We used multiple copies of the motif sequences to increase their repressive effect in the luciferase assay. The number five was chosen arbitrarily. The same repressive effect was observed when using three or seven copies of the motifs (Figure R3). Indeed, with the increase in motif copy number, such effects became more prominent. Copies of motifs were inserted into 5'UTR: 34-nt from 5' cap and 45-nt from the start codon. The gap between any two adjacent motif repeats was 4-nt. We have now added this information in the Materials and Methods sections.

Figure R3. Luciferase validation of candidate hexamer motifs with different number of motif copies. Using three and seven copies, we also observed the repressive effect of the two motifs in translation. Moreover, with the increase in motif copy number, such effects became more prominent. (n=3;

mean \pm SEM; n.s. $P > 0.05$, * $P < 0.05$, ** $P < 0.01$; student's *t*-test).

8. As it's well known that Kozak sequence plays an important role in efficient translation, I suggest to analyze the sequence data for the presence/absence of Kozak sequence. It would be interesting to know whether TSS isoform pairs varying in TE also differ in terms of Kozak sequence.

R: The consensus Kozak sequence is about 10-nt long and overlapping with the start codon. In our study, the two isoforms in comparison shared the same ORF. Therefore, there were no such cases where one isoform with the Kozak sequence and the other without.

9. How many isoforms have multiple translation regulatory sequence features in 5' UTR mentioned in Figure 6A? It is not explained what delta cumulative is representing here and how it is calculated, please describe it.

R: In the pairwise isoform comparison, 36.7% differed only in their 5'UTR length; In addition to 5'UTR length, 33.8% pairs differed only in one additional sequence feature, and 29.5% differed in at least two additional sequence features. 11

In analysis of combinatory contribution of sequence features, we sequentially added sequence features to models in the descending order of their individual contribution, and measured their cumulative contribution as the variance of observed TE divergence explained by the model combining these sequence features. Delta cumulative contribution was calculated as the additional variance explained by adding the specific sequencing feature to the combinatory models. Delta cumulative contribution was used to estimate the extent of additional information gained by considering one more feature given the interdependence between different features. In the revised manuscript, these terms have been clearly described in the figure legend and the Materials and Methods section.

10. In figure EV2 C and D the unit of Y-axis is not given. In fact, I doubt that translation efficiency calculated here can be compared with the ribosome footprinting data (Eichhorn et al, 2014) and proteomics data (Schwanhäusser et al, 2011). As these studies have different measures of TE and different methods were employed to calculate it. Moreover, in my understanding, for the ribosome footprinting Eichhorn et al used miRNA-inducible NIH3T3 cells, and to collect proteomics data, Schwanhäusser et al cultured NIH3T3 cells in SILAC medium which is different from current study. Different culture conditions and passage number may affect the result. So, in my opinion, statements related to this analysis should be removed from the manuscript.

R: In the revised manuscript, the figure labels for the Y-axis have been changed according to the suggestion from Reviewer #1. In Figure EV2C and D, we wanted to demonstrate the reasonable correlation between different approaches in measuring translational efficiency. As pointed out by the reviewer, this correlation may serve as a conservative estimate given the potential cellular difference. In previous publications for assessing the performance of similar approaches, such comparison has often been applied, with data generated from different laboratories (Spies *et al*, 2013; Floor & Doudna, 2016).

Minor Points:

1. Correct the grammatical errors, such as : Please change single quote (') to prime sign (′) throughout the manuscript. For instance, change 5' UTR to 5′ UTR. In Figure 2C Y-axis, correct spelling of Luciferase.

R: We thank the reviewer for pointing these out. We have corrected these typos accordingly.

2. In Figure EV1 B, please describe what Y-axis is and what different colors are representing.

R: We have added more description of Figure EV1B in the figure legend.

3. "Takahashi et al, 2012" reference is missing from the list.

R: We have added this reference in the revised version.

Reviewer #3:

Wang & Hou et al. show that variation in transcription start sites (TSSes) of mammalian transcripts (in cell culture) can affect their translational efficiency, as measured by their position on polysome profiles. By using CAGE-based techniques to define 5' ends of transcripts, the authors build on previous work to show that sequence elements and features in 5' leaders can impact downstream translation (*we will use 5' leader instead of 5' UTRs, since many 5' leaders are translated over their uORFs).*

As noted by the authors, this work is similar to recently-published work by Floor and Doudna (10.7554/eLife.10921), as well as Dieudonne et al. (10.1186/s12864-015-2179-8; not cited). However, a key distinction is that Wang & Hou et al. use experimental techniques to identify 5' ends de novo and quantify their use, rather than relying on the quantification of 5' ends of existing annotations from RNA-Seq data. The authors do show that this de novo annotation and quantification is important, as many of these identified 5' ends do not coincide with annotated 5' ends.

All things considered, we recommend this manuscript for publication. Overall, the work is well-documented, and high throughput experiments are independently validated. While we would like to see some analyses more explicitly explained, we do not see the need for further experiments. The findings of this paper are no surprise to anyone in the field, but have not been rigorously demonstrated prior to this work, particularly in mammalian cells. The work would be a meaningful addition to the existing literature.

R: We thank the reviewer for the positive comments.

Major points:

1. Please define exactly what is meant by the "gross 5'-UTR region." This is an important concept and does not appear to be defined in either the results or the methods sections.

R: For each gene, the gross 5'UTR region includes its annotated 5'UTRs (from the most 5' annotated TSS to the most 3' annotated start codon) and 1-kb upstream of the most 5' annotated TSS. We have made this notation clearer in the revised manuscript.

2. In-frame upstream AUGs should be subdivided into AUGs that are merely N-terminal extensions of the CDS, and bona fide uORFs that may be in-frame with, but terminate before the CDS (Figs 4B, C). This would help determine why in-frame uAUGs differ from out-of-frame uAUGs in their effects on downstream translation.

R: In this study, we grouped the AUGs in 5'UTR into upstream ORFs and upstream AUGs according to whether the AUGs have in-frame stop codons in the 5'UTR. Therefore in this manuscript, upstream AUGs do not include uORFs.

3. It is also not clear what "individual", "cumulative" and "delta cumulative" mean in Fig 6A: please interpret this for the reader (who should not be expected to fire-up the R 'earth' package to do so).

R: To assess the individual contribution of sequence features, we built quantitative models for each feature separately, and took the variance of observed TE divergence explained by each model as their individual contribution. In the analysis of combinatory contribution of sequence features, we sequentially added sequence features to models in the descending order of their individual contribution, and measured their cumulative contribution as the variance explained by the model combining these sequence features. Delta cumulative contribution was calculated as the additional variance explained by adding the specific sequencing feature to the combinatory models. Delta cumulative contribution was used to estimate the extent of additional information gained by considering one more feature given the interdependence between different features.

In the revised manuscript, these terms have been clearly described in the Materials and Methods

sections and the figure legend.

4. Please justify the use of the MARS approach instead of simpler linear models (with appropriately transformed parameters). Similarly, the specific parameters used should be enumerated more clearly.

R: MARS is a non-parametric regression approach that allows modeling the nonlinear relationship between sequence features and the resulted TE changes. In contrast, linear models cannot directly capture non-linearity, and thus are in general with worse performance. Although proper transformation can deal with non-linearity, it requires prior knowledge, such as what transformation and which parameters are more appropriate. In comparison, there is no such requirement for MARS modeling, which is able to reconstruct the functional relationship solely from the data. The MARS approach has been used to address similar modeling problems, as exemplified in (Vogel *et al*, 2010). The parameters used for MARS modeling with R package 'earth' were 'degree=1, penalty=2, thresh=0.001, fast.k=0, fast.beta=0'. The parameter 'degree' defines the maximum degree of interaction between variables, and the value 1 means to build additive models with no interaction terms allowed. The parameter 'penalty' is the penalty in generalized cross validation, and the value 2 is the default setting for degree=1. The setting 'thresh=0.001' is one of the computation termination criteria, tuning between computing time and model performance. Setting 'fast.k=0, fast.beta=0' disables fast calculation. In the revised manuscript, we have added the above information in the Materials and Methods section.

5. Cycloheximide pre-treatment prior to cell lysis (as was done in the experiments in this work) causes a known experimental artifact: because it stalls elongating ribosomes, it tends to cause a 5' pile-up of ribosomes, which can be observed in ribosome profiling data. This artifact can potentially confound polysome profiling data, as shorter ORFs may be predicted to have disproportionately more ribosomes (per unit length), as would polycistronic transcripts (i.e. transcripts with multiple uORFs). Unfortunately, the established protocols for polysome profiling do not account for this. We recommend checking if this is an issue, by plotting calculated associated ribosomes against the length of the CDS, perhaps for the transcripts without uORFs / uAUGs in the 5' leader. If 5' pile-up were an issue, one might expect a significant residual number of associated ribosomes as the length of the CDS goes to 0. Should this y-intercept be substantial, it would need to be accounted for in the calculation of translational efficiency.

R: Following the reviewer's suggestion, we plotted ribosome numbers against CDS length for each TSS isoform without uORFs/uAUGs. As shown in Figure R4A, we found overall a non-linear relationship between ribosome number and CDS length. To estimate the Y-intercept with linear regression, we had to restrict the analysis in a subset of transcripts with relatively short CDS length. For this purpose, we performed a series of tests for non-linearity (White test) with restricted CDS length (i.e. from 0-nt to a variety of upper bounds) and found that the linear relationship maintained until the upper bound exceeded 800-nt (Figure R4B). Thereafter, the regression analysis was performed on transcripts with CDS length less than 800-nt, which resulted in a fitted line with Y-intercept at 0.515 (Figure R4A), indicating an insubstantial effect of cycloheximide treatment, if any.

Nevertheless, we further examined how such potential effect could affect our result by checking the difference between with and without considering the 'additional' ribosomes in calculating TE fold-changes between isoform-pairs. As shown in Figure R4C, if we deducted 0.5 ribosome from the original ribosome numbers we set for the monosome and polysome fractions (from {0,0,1,2.5,4.5,7.5,12} to {0,0,0.5,2,4,7,11.5}), the newly calculated values almost stayed the same. Even if increasing the deducted ribosome number from 0.5 to 1 (from {0,0,1,2.5,4.5,7.5,12} to {0,0,0,1.5,3.5,6.5,11}), as shown in Figure R4D, the new values still correlate well with the previously reported ones. Collectively, even if there might be small effect of cycloheximide treatment, the artifact would not affect our results in this study.

Figure R4. Investigation of potential effect of cycloheximide treatment in polysome profiling. (A) scatterplot showing the nonlinear relationship between per-mRNA ribosome numbers and ORF length, so that the linear fitting was only based on points with ORF length less than 800-nt. (B) Plot showing adjusted P-values (Y-axis) of a series of non-linearity tests (White test) for points in (A) with restricted ORF length, i.e. from 0-nt to a variety of upper bounds shown in the X-axis. (C,D) Scatterplots comparing \log_2 TE divergence calculated with (Y-axis) and without (X-axis) considering the possible 'additional' ribosomes caused by cycloheximide treatment.

6. Gene Ontology enrichment analyses based on sequencing data are necessarily biased by gene expression levels, as highly expressed genes have more reads and so pass statistical thresholds more readily. This problem can be ameliorated by using a method such as G_Oseq (PMID 20132535). The authors should repeat their enrichment analysis using an appropriate bias correction.

R: We thank the reviewer for this suggestion. Indeed, we observed that genes with low expression showed a slight tendency to have only one TSS detected. Following the suggestion, we repeated the GO analysis using GSeq (Young *et al*, 2010) (version 1.22.0) in R, and the enrichment result for multi-TSS genes vs. expressed genes was similar to what we previously reported. However, the result for genes with isoform-divergent TE vs. multi-TSS genes became non-significant. In the revised manuscript, we have adjusted/removed the results according to the GSeq analysis.

7. The sequencing data generated in this study must be deposited in a public repository (e.g. SRA or ENA).

R: The raw sequencing data has been submitted to NCBI GEO database under accession number GSE78241. This information has been added in the revised manuscript. For review purpose, the record is accessible through the link

<http://www.ncbi.nlm.nih.gov/geo/query/acc.cgi?token=izetoewufxijriz&acc=GSE78241>

Minor points:

1. Spearman's rank correlation coefficient is denoted by the Greek letter "rho," not "R."

R: We have changed the notation accordingly.

2. It would be useful to show the distribution and dynamic range of translational efficiency observed across various genes, particularly as a comparison to similar measures of translational efficiency typically calculated from ribosome profiling data.

R: According to the suggestion from the reviewer, we have added histograms in Figures EV2C and D to show the distribution and dynamic ranges of TE measured by different approaches. We have also redrawn the scatterplots in log scale for both axes.

3. While they are likely to be well-correlated with each other in most cases, we suggest the use of ensemble free energy (EFE) over minimum free energy (EFE) as the most relevant value for assessing secondary structure stability. This can be evaluated using ViennaRNA's *pf_fold* function, or *RNAfold -p* (Figs 5A, B).

R: We thank the reviewer for the suggestion. Based on EFE, our conclusion remained the same (Figure EV6C and EV6D). In the revised manuscript, we have added this analysis in Materials and Methods, and the Results section.

4. Technically, ORFs are defined purely by sequence: the transcript segment defined by a start codon on the 5' end, and an in-frame stop codon on the 3' end. Thereafter, uORFs can be classified (by some threshold) as translated or untranslated. This distinction should be made for Figs 4D-G.

R: Thanks for the suggestion. We modified the figure labels and legends accordingly.

5. Given that arbitrary thresholds and significance levels are used to classify genes as multiply-initiated (4153/9951) or translationally-divergent (745/4153), we would qualify such statements appropriately, e.g. 4,153 of 9,951 expressed genes showed significant initiation at multiple TSSs.

R: We thank the reviewer for this suggestion, and we have edited the sentence accordingly.

6. Scatter plots are over-plotted; use some transparency in the points so that the density at different regions is better appreciated.

R: As suggested by the reviewer, we have adjusted the scatterplots in Figures 2A, 3B, and 6B in the revised manuscript.

7. The genes that were individually validated in Fig 2C should be highlighted accordingly on Fig 6B. They could be depicted as horizontal line segments, where both observed values (from individual experiments and high-throughput data) are joined.

R: As suggested by the reviewer, we added the plot as Figure EV7. In this figure, we excluded two genes, *Eif1ad* and *Ndufb11*, as they were not included in the analysis of sequence features, due to

alternative splicing between their proximal and distal TSSs.

8. Fig 3C is redundant (all the information is already captured in Fig 3A).

R: In the revised manuscript, we have removed this panel.

9. While this may not be necessary for publication, we highly encourage the publication and/or public deposition of all code and data tables necessary for the analyses presented in this work.

R: As suggested by the reviewer, we have made the analysis scripts and processed data sets available on-line at <https://github.com/sunlightwang/CAPTRE>, and this information has been added in the revised manuscript.

Citation issues: 1. More recent work quantifying the relative contribution of transcription and translation should be cited as well: 10.1126/science.1259038 and 10.7717/peerj.270, particularly because they suggest that transcriptional control plays a larger contribution (see 10.1126/science.aaa8332)

2. Another key prior work that has investigated the contribution of isoforms to translation (albeit in yeast) is 10.1038/nature12121. It should be referenced also. 3. 10.1186/s12859-014-0380-4 could be a useful citation for explaining the lack of effect in non-AUG codons initiation.

R: We appreciate the reviewer for pointing out the important references for our manuscript. In the revised manuscript, we have added all abovementioned references.

REFERENCES

- Carrieri C, Cimatti L, Biagioli M & Beugnet A (2012) Long non-coding antisense RNA controls Uchl1 translation through an embedded SINEB2 repeat. *Nature* 491: 454–457
- Farh KK, Grimson A, Jan C, Lewis BP, Johnston WK, Lim LP, Burge CB & Bartel DP (2005) The widespread impact of mammalian MicroRNAs on mRNA repression and evolution. *Science* (80-.). 310: 1817–1821
- Floor SN & Doudna JA (2016) Tunable protein synthesis by transcript isoforms in human cells. *Elife* 5: e10921
- Lewis BP, Shih IH, Jones-Rhoades MW, Bartel DP & Burge CB (2003) Prediction of mammalian microRNA targets. *Cell* 115: 787–798
- Lim LP, Lau NC, Garrett-Engle P, Grimson A, Schelter JM, Castle J, Bartel DP, Linsley PS & Johnson JM (2005) Microarray analysis shows that some microRNAs downregulate large numbers of target mRNAs. *Nature* 433: 769–73
- Mokrejs M, Masek T, Vopálenský V, Hlubucek P, Delbos P & Pospisek M (2010) IRESite--a tool for the examination of viral and cellular internal ribosome entry sites. *Nucleic Acids Res.* 38: D131–6
- Ørom UA, Nielsen FC & Lund AH (2008) MicroRNA-10a binds the 5'UTR of ribosomal protein mRNAs and enhances their translation. *Mol. Cell* 30: 460–71
- Spies N, Burge CB & Bartel DP (2013) 3' UTR-isoform choice has limited influence on the stability and translational efficiency of most mRNAs in mouse fibroblasts. *Genome Res.* 23: 2078–2090
- Vogel C, Abreu RDS & Ko D (2010) Sequence signatures and mRNA concentration can explain two-thirds of protein abundance variation in a human cell line. *Mol. Syst. Biol.* 6: 400
- Young MD, Wakefield MJ, Smyth GK & Oshlack A (2010) Gene ontology analysis for RNA-seq: accounting for selection bias. *Genome Biol* 11: R14
- Zhao Y, Li H, Fang S, Kang Y, Wu W, Hao Y, Li Z, Bu D, Sun N, Zhang MQ & Chen R (2016)

NONCODE 2016: an informative and valuable data source of long non-coding RNAs. *Nucleic Acids Res.* 44: D203–8 19

Zucchelli S, Fasolo F, Russo R, Cimatti L, Patrucco L, Takahashi H, Jones MH, Santoro C, Sblattero D, Cotella D, Persichetti F, Carninci P & Gustincich S (2015) SINEUPs are modular antisense long non-coding RNAs that increase synthesis of target proteins in cells. *Front. Cell. Neurosci.* 9: 174

Corresponding Author Name: Wei Chen
Journal Submitted to: Molecular Systems Biology
Manuscript Number: MSB-16-6941